# ASGNN: Graph Neural Networks with Adaptive Structure

## Abstract

The graph neural network (GNN) models have presented impressive achievements in numerous machine learning tasks. However, many existing GNN models are shown to be vulnerable to adversarial attacks, which creates a stringent need to build robust GNN architectures. In this work, we propose a novel interpretable message passing scheme with adaptive structure (ASMP) to defend against adversarial attacks on graph structure. Layers in ASMP are derived based on optimization steps that minimize an objective function that learns the node feature and the graph structure simultaneously. ASMP is adaptive in the sense that the message passing process in different layers is able to be carried out over dynamically adjusted graphs. Such property allows more fine-grained handling of the noisy (or perturbed) graph structure and hence improves the robustness. Convergence properties of the ASMP scheme are theoretically established. Integrating ASMP with neural networks can lead to a new family of GNN models with adaptive structure (ASGNN). Extensive experiments on semi-supervised node classification tasks demonstrate that the proposed ASGNN outperforms the state-of-the-art GNN architectures in terms of classification performance under various adversarial attacks.

## 1 Introduction

Graphs, or networks, are ubiquitous data structures in many fields of science and engineering (Newman, 2018), like molecular biology, computer vision, social science, financial technology, etc. In the past few years, due to its appealing capability of learning representations through message passing over the graph structure, graph neural network (GNN) models have become popular choices for processing graph-structured data and have achieved astonishing success in various applications (Kipf and Welling, 2017; Bronstein et al., 2017; Wu et al., 2020; Zhou et al., 2020; Wu et al., 2022). However, existing GNN backbones such as the graph convolutional network (GCN) (Kipf and Welling, 2017) and the graph attention network (Veličković et al., 2018) are shown to be extremely vulnerable to carefully designed adversarial attacks on the graph structure (Sun et al., 2018; Jin et al., 2021; Günnemann, 2022). With unnoticeable malicious manipulations of the graph, the performance of GNNs significantly drops and may even be worse than the performance of a simple baseline that ignores all the relational information among data feature (Dai et al., 2018; Zügner et al., 2018; Zügner and Günnemann, 2019; Zhang and Zitnik, 2020). With the increasing deployments of GNN models in various real-world applications, it is of vital importance to ensure their reliability and robustness, especially in scenarios, such as medical diagnosis and credit scoring, where a deflected model can lead to dramatic consequences (Günnemann, 2022).

To improve the robustness of GNNs with a potentially noisy graph structure input, a natural idea is to "purify" the given graph structure. Existing work in this line can be roughly classified into two categories. The first category of robustifying GNNs can be viewed as a two-stage approach (Wu et al., 2019b; Entezari et al., 2020; Gu et al., 2021). A purified graph is firstly obtained by "pre-processing" the input graph structure leveraging on information from the node feature. Next, a GNN model is trained based on this purified graph. For example, in the GNN-Jaccard method (Wu et al., 2019b), a new graph is obtained by removing the edges with small "Jaccard similarity." In Entezari et al. (2020), observing that adversarial attacks can scale up the rank of the graph adjacency matrix, the authors propose to use a low-rank approximation version of the given graph adjacency matrix as a substitute. In the second category, the graph adjacency matrix in a GNN model is treated as an unknown, a purified graph structure with a parameterized form will be "learned" through optimizing

the supervised GNN training loss (Zhu et al., 2022). For example, in Franceschi et al. (2019), the graph adjacency matrix is directly learned with a GNN in a bilevel optimization way, where a full parametrization of the graph adjacency matrix is adopted. Moreover, under this full parametrization setting, structural regularizers are adopted in Jin et al. (2020); Luo et al. (2021) as augmentations on the training loss function to promote certain properties of the purified graph. Besides the full parametrization approach, a multi-head weighted cosine similarity metric function (Chen et al., 2020) and a GNN model (Yu et al., 2020) have also been used to parameterize the graph adjacency matrix for structure learning.

Going beyond purifying the graph structures to robustify the GNN models, there are also efforts on designing robust GNN architectures via directly designing the feature aggregation schemes. Under the observation that aggregation functions such as sum, weighted mean, or the max operations can be arbitrarily distorted by only a single outlier node, Geisler et al. (2020); Wang et al. (2020); Zhang and Lu (2020) try to design robust GNN models via designing robust aggregation functions. Moreover, some works apply the attention mechanism (Veličković et al., 2018) to mitigate the influence of adversarial perturbations. For example, Zhu et al. (2019) consider the node feature following a Gaussian distribution and use the variance information to determine the attention scores. Tang et al. (2020) use clean graph information and their adversarial counterparts to train an attention mechanism to learn to assign small attention scores to the perturbed edges. In Zhang and Zitnik (2020), the authors define an attention mechanism based on the similarity of neighboring nodes.

Different from existing approaches to robustify GNNs, in this work, we propose a novel robust and interpretable message passing scheme with adaptive structure (ASMP). Based on ASMP, a family of GNN models with adaptive structure (ASGNN) can be designed. Prior works have revealed that the message passing processes in a class of GNNs are actually (unrolled) gradient steps for solving a graph signal denoising (GSD) problem (Zhu et al., 2021; Ma et al., 2021; Zhang and Zhao, 2022). ASMP is actually generated by an alternating (proximal) gradient descent algorithm for simultaneously denoising the graph signal and the graph structure. Designed in such a principled way, ASMP is not only friendly to back-propagation training but also achieves the desired structure adaptivity with a theoretical convergence guarantee. Once trained, ASMP can be naturally interpreted as a parameter-optimized iterative algorithm. This work falls into the category of GNN architecture designs. Conceptually different from the existing robustified GNNs with *fixed* graph structure, ASGNN interweaves the graph purification process and the message passing process, which makes it possible to conduct message passing over different graph structures at different layers, i.e., in an *adaptive* graph structure fashion. Thus, an edge might be excluded in some layers but included in other layers, depending on the dynamic structure learning process. Such property allows more fine-grained handling of perturbations than existing graph purification methods that use a single graph in the entire GNN. To be more specific, the major contributions of this work are highlighted in the following.

- We propose a novel message passing scheme over graphs called ASMP with convergence guarantee and specifications. To the best of our knowledge, ASMP is the first message passing scheme with adaptive structure that is designed based on an optimization problem.

- Based on ASMP, a family of GNN models with adaptive structure, named ASGNN, are further introduced. The adaptive structure in ASGNN allows more fine-grained handling of noisy graph structures and strengthens the model robustness against adversarial attacks.

- Extensive experiments under various adversarial attack scenarios showcase the superiority of the proposed ASGNN. The numerical results corroborate that the adaptive structure property inherited in ASGNN can help mitigate the impact of perturbed graph structure.

## 2 PRELIMINARIES AND BACKGROUND

An unweighted graph with self-loops is denoted as $\mathcal{G} = (\mathcal{V}, \mathcal{E})$, where $\mathcal{V}$ and $\mathcal{E}$ denote the node set and the edge set, respectively. The graph adjacency matrix is given by $\mathbf{A} \in \mathbb{R}^{N \times N}$. We denote by $\mathbf{1}$ and $\mathbf{I}$ the all-one column vector and the identity matrix, respectively. Given $\mathbf{D} = \mathrm{Diag}\,(\mathbf{A1}) \in \mathbb{R}^{N \times N}$ as the diagonal degree matrix, the Laplacian matrix is defined as $\mathbf{L} = \mathbf{D} - \mathbf{A}$. We denote by $\mathbf{A}_{\mathrm{rw}} = \mathbf{D}^{-1}\mathbf{A}$ the random walk (or row-wise) normalized adjacency matrix and by $\mathbf{A}_{\mathrm{sym}} = \mathbf{D}^{-\frac{1}{2}}\mathbf{A}\mathbf{D}^{-\frac{1}{2}}$ the symmetric normalized adjacency matrix. Subsequently, the random walk normalized and symmetric

normalized Laplacian matrices are defined as $\mathbf{L}_{\mathrm{rw}} = \mathbf{I} - \mathbf{D}^{-1}\mathbf{A}$ and $\mathbf{L}_{\mathrm{sym}} = \mathbf{I} - \mathbf{D}^{-\frac{1}{2}}\mathbf{A}\mathbf{D}^{-\frac{1}{2}}$, respectively. $\mathbf{X} \in \mathbb{R}^{N \times M}$ ($M$ is assumed to be the dimension of the node feature) is a node feature matrix or a graph signal, and its $i$-th row $\mathbf{X}_{i,:}$ represents the feature vector at the $i$-th node with $i = 1, \ldots, N$. $\mathbf{X}_{ij}$ (or $[\mathbf{X}]_{ij}$) denotes the $(i, j)$-th element of $\mathbf{X}$ with $i, j = 1, \ldots, N$. For vector $\mathbf{X}_{i,:}$, $\mathbf{X}_{i,:}^{-1}$ represents its element-wise inverse.

## 2.1 GNNs as Graph Signal Denoising

In the literature (Yang et al., 2021; Pan et al., 2021; Zhu et al., 2021), it has been realized that the message passing layers for feature learning in many GNN models could be uniformly interpreted as gradient steps for minimizing certain energy functions, which carries a meaning of GSD (Ma et al., 2021). Recently, Zhang and Zhao (2022) further showed that some popular GNNs are neural networks induced from unrolling (proximal) gradient descent algorithms for solving specific GSD problems. Taking the approximate personalized propagation of neural predictions (APPNP) model (Klicpera et al., 2019) as an example, the initial node feature matrix $\mathbf{Z}$ is first pre-propcessed by a multilayer perceptron $g_\theta(\cdot)$ with model parameter $\boldsymbol{\theta}$ producing an output $\mathbf{X} = g_\theta(\mathbf{Z})$, and then $\mathbf{X}$ is fed into a $K$-layer message passing scheme given as follows:

$$\mathbf{H}^{(0)} = \mathbf{X}, \quad \mathbf{H}^{(k+1)} = (1 - \alpha)\,\mathbf{A}_{\mathrm{sym}}\mathbf{H}^{(k)} + \alpha\mathbf{X}, \ \text{ for } k = 0, \ldots, K - 1, \tag{1}$$

where $\mathbf{H}^{(0)}$ denotes the input feature of the message passing process, $\mathbf{H}^{(k)}$ represents the learned feature after the $k$-th layer, and $\alpha$ is the teleport probability. Therefore, the message passing of an APPNP model is fully specified by two parameters, namely, a graph structure matrix $\mathbf{A}_{\mathrm{sym}}$ and a parameter $\alpha$, in which $\mathbf{A}_{\mathrm{sym}}$ assumes to be known beforehand and $\alpha$ is treated as a hyperparameter.

From an optimization perspective, the message passing process in Eq. (1) can be seen as executing $K$ steps of gradient descent to solve a GSD problem with initialization $\mathbf{H}^{(0)} = \mathbf{X}$ and step size $0.5$ (Zhu et al., 2021; Ma et al., 2021; Zhang and Zhao, 2022), which is given by

$$\underset{\mathbf{H} \in \mathbb{R}^{N \times M}}{\text{minimize}} \quad \alpha \|\mathbf{H} - \mathbf{X}\|_{\mathrm{F}}^2 + (1 - \alpha)\,\mathrm{Tr}\!\left(\mathbf{H}^\top \mathbf{L}_{\mathrm{sym}}\mathbf{H}\right), \tag{2}$$

where $\mathbf{X}$ and $\alpha$ are given and share the same meaning as in Eq. (1). In Problem (2), the first term is a fidelity term forcing the recovered graph signal $\mathbf{H}$ to be as close as possible to a noisy graph signal $\mathbf{X}$, and the second term is the symmetric normalized Laplacian smoothing term measuring the variation of the graph signal $\mathbf{H}$, which can be explicitly expressed as

$$\mathrm{Tr}\left(\mathbf{H}^\top \mathbf{L}_{\mathrm{sym}}\mathbf{H}\right) = \frac{1}{2}\sum_{i=1}^{N}\sum_{j=1}^{N}\mathbf{A}_{ij}\left\|\frac{\mathbf{H}_{i,:}}{\sqrt{\mathbf{D}_{ii}}} - \frac{\mathbf{H}_{j,:}}{\sqrt{\mathbf{D}_{jj}}}\right\|_2^2. \tag{3}$$

For more technical discussions on relationships between GNNs with iterative optimization algorithms for solving GSD problems, please refer to Ma et al. (2021); Zhang and Zhao (2022). Apart from using the lens of optimization to interpret existing GNN models, there are also literature (Liu et al., 2021b; Chen et al., 2021; Fu et al., 2022) working on building new GNN architectures based on designing novel optimization problems and the corresponding iterative algorithms (more discussions are provided in Appendix A).

## 2.2 Graph Learning with Structural Regularizers

Structural regularizers are commonly adopted to promote certain desirable properties when learning a graph (Kalofolias, 2016; Pu et al., 2021). In the following, we discuss several widely used graph structural regularizers which will be incorporated into the design of ASMP. We denote the learnable graph adjacency matrix as $\mathbf{S}$ satisfying $\mathbf{S} \in \mathcal{S}$, where

$$\mathcal{S} = \left\{\mathbf{S} \in \mathbb{R}^{N \times N} \mid 0 \leq \mathbf{S}_{ij} \leq 1, \text{ for } i, j = 1, \ldots, N\right\}$$

defines the class of adjacency matrices. Under the assumption that node feature changes smoothly between adjacent nodes (Ortega et al., 2018), the Laplacian smoothing regularization term is commonly considered in graph structure learning. Eq. (3) is the symmetric normalized Laplacian smoothing term, and a random walk normalized alternative can be similarly defined by replacing $\mathbf{L}_{\mathrm{sym}}$ in Eq. (3) by $\mathbf{L}_{\mathrm{rw}}$.

Real-world graphs are normally sparsely connected, which can be represented by sparse adjacency matrices. Moreover, it is also observed that singular values of these adjacency matrices are commonly small (Zhou et al., 2013; Kumar et al., 2020). However, a noisy adjacency matrix (e.g., one perturbed by adversarial attacks) tends to be dense and to gain singular values in larger magnitudes (Jin et al., 2020). In view of this, graph structural regularizers for promoting sparsity and/or suppressing the singular values are widely adopted in the literature of graph learning (Kalofolias, 2016; Egilmez et al., 2017; Dong et al., 2019). Specifically, the $\ell_1$-norm of the adjacency matrix is often used to promote sparsity, defined as $\|\mathbf{S}\|_1 = \sum_{i,j=1}^N |\mathbf{S}_{ij}|$. For penalizing the singular values, the $\ell_1$-norm and the $\ell_2$-norm on the singular value vector of the adjacency matrix $\mathbf{S}$ can help. Equivalently, they can be translated to be the nuclear norm and the Frobenius norm on $\mathbf{S}$, which are given by $\|\mathbf{S}\|_* = \sum_{i=1}^N \sigma_i(\mathbf{S})$ and $\|\mathbf{S}\|_F = \sqrt{\sum_{i=1}^N \sigma_i^2(\mathbf{S})}$, respectively, where $\sigma_1(\mathbf{S}) \geq \cdots \geq \sigma_N(\mathbf{S})$ denote the ordered singular values of $\mathbf{S}$. These two regularizers both restrict the scale of the singular values while the nuclear norm also promotes low-rankness. A recent study (Deng et al., 2022) points out that graph learning methods with low-rank promoting regularizers may lose a wide range of spectrum of the clean graph corresponding to important structure in the spatial domain. Thus, the nuclear norm regularizer may impair the quality of the reconstructed graph and therefore limit the performance of GNNs. Besides, the nuclear norm is not amicable for back-propagation and incurs high computational complexity (Luo et al., 2021). Arguably, the Frobenius norm of $\mathbf{S}$ is a more suitable regularizer for graph structure learning in comparison with the nuclear norm.

## 3 THE PROPOSED GRAPH NEURAL NETWORKS

In this section, we first motivate the design principle based on jointly node feature learning and graph structure learning. Then, we develop an efficient optimization algorithm for solving this optimization problem, which eventually leads to a novel message passing scheme with adaptive structure (ASMP). After that, we provide interpretations, convergence guarantees, and specifications of ASMP. Finally, integrating ASMP with deep neural networks ends up with a new family of GNNs with adaptive structure, named ASGNNs.

### 3.1 A NOVEL DESIGN PRINCIPLE WITH ADAPTIVE GRAPH STRUCTURE

As discussed in Section 2.1, the message passing procedure in many popular GNNs can be viewed as performing graph signal denoising (or node feature learning) (Zhu et al., 2021; Ma et al., 2021; Pan et al., 2021; Zhang and Zhao, 2022) over a prefixed graph. Unfortunately, if some edges in the graph are task-irrelevant or even maliciously manipulated, the node feature learned may not be appropriate for the downstream tasks. Motivated by this, we propose a new design principle for message passing, that is, to learn the node feature and the graph structure simultaneously. It enables learning an adaptive graph structure from the feature for the message passing procedure. Hence, such a message passing scheme can potentially improve robustness against noisy input graph structure.

Specifically, we construct an optimization objective by augmenting the GSD objective in Eq. (2) (we have used a random walk normalized graph Laplacian smoothing term) with a structure fidelity term $\|\mathbf{S} - \mathbf{A}\|_F^2$, where $\mathbf{A}$ is the given initial graph adjacency matrix, and the structural regularizers $\|\mathbf{S}\|_1$ and $\|\mathbf{S}\|_F^2$. Then we obtain the following optimization problem:

$$\underset{\mathbf{H} \in \mathbb{R}^{N \times M}, \, \mathbf{S} \in \mathcal{S}}{\text{minimize}} \; p(\mathbf{H}, \mathbf{S}) = \overbrace{\|\mathbf{H} - \mathbf{X}\|_F^2 + \lambda \mathrm{Tr}(\mathbf{H}^\top \mathbf{L}_{\mathrm{rw}} \mathbf{H})}^{\text{feature learning}} + \underbrace{\gamma \|\mathbf{S} - \mathbf{A}\|_F^2 + \mu_1 \|\mathbf{S}\|_1 + \mu_2 \|\mathbf{S}\|_F^2}_{\text{structure learning}},$$

(4)

where $\mathbf{H}$ is the feature variable, $\mathbf{S}$ is the structure variable, and $\gamma$, $\lambda$, $\mu_1$, and $\mu_2$ are parameters balancing different terms. To enable the interplay between feature learning and structure learning, the Laplacian smoothing term is concerned with $\mathbf{S}$ rather than $\mathbf{A}$, i.e., $\mathbf{L}_{\mathrm{rw}} = \mathbf{I} - \mathbf{D}^{-1}\mathbf{S}$ with $\mathbf{D} = \mathrm{Diag}(\mathbf{S1})$. When adversarial attacks exist, a perturbed adjacency matrix $\mathbf{A}$ will be generated. Since attacks are generally designed to be unnoticeable (Jin et al., 2021), the perturbed graph adjacency matrix is largely similar to the original graph matrix in value. In view of this, we also

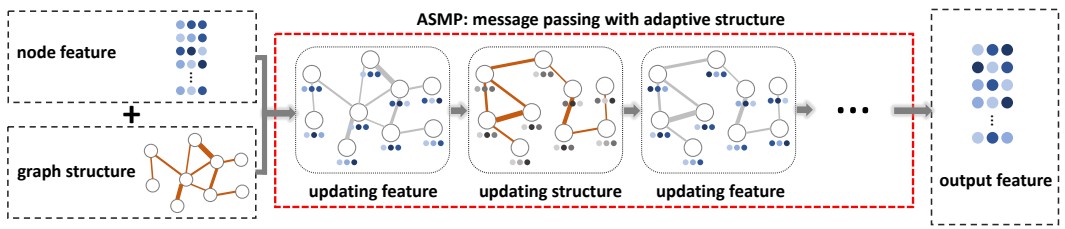

Figure 1: Illustration of ASMP. (ASMP takes the node feature matrix $\mathbf{X}$ and graph structure matrix $\mathbf{A}$ as input. Different colors of the feature indicate different embedding values. Different width of the edges indicate different weight values. ASMP updates the node feature and the graph structure in an alternating way.)

include a structural fidelity term $\|\mathbf{S} - \mathbf{A}\|_{\mathrm{F}}^2$. The motivation for introducing the last two regularizers has been elaborated in Section 2.2.

### 3.2   ASMP: MESSAGE PASSING WITH ADAPTIVE STRUCTURE

Following the idea that the message passing of a GNN model can be derived based on the optimization of a GSD objective function (Ma et al., 2021; Zhang and Zhao, 2022), we can obtain a message passing scheme from Problem (4). Different from the existing GSD problems for GNN model design with only the feature variable, Problem (4) is nonconvex and much more challenging. To obtain an efficient iterative algorithm that is friendly to back-propagation training, we propose to use the alternating (proximal) gradient descent method (Parikh and Boyd, 2014), i.e., alternatingly optimizing one variable by taking one (proximal) gradient step at a time with the other variable fixed. (Note that a joint optimization approach is also eligible, while it would lead to slower convergence than the alternating optimization approach. More details can be found in Appendix D.)

We denote by $\mathbf{H}^{(k)}$ and $\mathbf{S}^{(k)}$ the variables at the $k$-th iteration ($k = 0, \ldots, K$). In the following, the update rules for $\mathbf{H}$ and $\mathbf{S}$ will be discussed, respectively.

**Updating node feature matrix $\mathbf{H}$**: Given $\{\mathbf{H}^{(k)}, \mathbf{S}^{(k)}\}$, the subproblem with respect to feature matrix $\mathbf{H}$ is given by

$$\underset{\mathbf{H} \in \mathbb{R}^{N \times M}}{\text{minimize}} \quad \|\mathbf{H} - \mathbf{X}\|_{\mathrm{F}}^2 + \lambda \mathrm{Tr}\left(\mathbf{H}^\top \mathbf{L}_{\mathrm{rw}}^{(k)} \mathbf{H}\right), \tag{5}$$

where $\mathbf{L}_{\mathrm{rw}}^{(k)} = \mathbf{I} - \mathrm{Diag}\left(\mathbf{S}^{(k)}\mathbf{1}\right)^{-1}\mathbf{S}^{(k)}$. One gradient step for $\mathbf{H}$ is computed as

$$\begin{aligned}
\mathbf{H}^{(k+1)} &= \mathbf{H}^{(k)} - \eta_1 \left(2\mathbf{H}^{(k)} - 2\mathbf{X} + 2\lambda \mathbf{L}_{\mathrm{rw}}^{(k)} \mathbf{H}^{(k)}\right) \\
&= \mathbf{H}^{(k)} - \eta_1 \left(2\mathbf{H}^{(k)} - 2\mathbf{X} + 2\lambda \left(\mathbf{I} - \mathrm{Diag}\left(\mathbf{S}^{(k)}\mathbf{1}\right)^{-1}\mathbf{S}^{(k)}\right)\mathbf{H}^{(k)}\right) \\
&= (1 - 2\eta_1 - 2\eta_1\lambda)\mathbf{H}^{(k)} + 2\eta_1\lambda\mathrm{Diag}\left(\mathbf{S}^{(k)}\mathbf{1}\right)^{-1}\mathbf{S}^{(k)}\mathbf{H}^{(k)} + 2\eta_1\mathbf{X},
\end{aligned}$$

where $\eta_1$ denotes the step size.

**Updating graph structure matrix $\mathbf{S}$**: Given $\{\mathbf{H}^{(k+1)}, \mathbf{S}^{(k)}\}$ and $\mathrm{Tr}(\mathbf{H}^{(k+1)\top}\mathbf{L}_{\mathrm{rw}}\mathbf{H}^{(k+1)}) = \mathrm{Tr}(\mathbf{H}^{(k+1)\top}\mathbf{H}^{(k+1)}) - \mathrm{Tr}(\mathbf{H}^{(k+1)\top}\mathrm{Diag}(\mathbf{S}\mathbf{1})^{-1}\mathbf{S}\mathbf{H}^{(k+1)})$, the subproblem for $\mathbf{S}$ becomes

$$\underset{\mathbf{S} \in \mathcal{S}}{\text{minimize}} \quad \gamma\|\mathbf{S} - \mathbf{A}\|_{\mathrm{F}}^2 - \lambda\mathrm{Tr}\left(\mathbf{H}^{(k+1)\top}\mathrm{Diag}(\mathbf{S}\mathbf{1})^{-1}\mathbf{S}\mathbf{H}^{(k+1)}\right) + \mu_1\|\mathbf{S}\|_1 + \mu_2\|\mathbf{S}\|_{\mathrm{F}}^2. \tag{6}$$

Due to the non-smoothness of the objective function, we apply one step of the proximal gradient descent (Parikh and Boyd, 2014) for this problem. Define

$$\begin{aligned}
\mathbf{T}^{(k)} = {}& (2\gamma + 2\mu_2)\,\mathbf{S}^{(k)} - 2\gamma\mathbf{A} - \lambda\mathrm{Diag}\left(\mathbf{S}^{(k)}\mathbf{1}\right)^{-1}\mathbf{H}^{(k+1)}(\mathbf{H}^{(k+1)})^\top \\
&+ \lambda\mathrm{Diag}\left(\mathrm{Diag}\left(\mathbf{S}^{(k)}\mathbf{1}\right)^{-1}\mathbf{S}^{(k)}\mathbf{H}^{(k+1)}(\mathbf{H}^{(k+1)})^\top\mathrm{Diag}\left(\mathbf{S}^{(k)}\mathbf{1}\right)^{-1}\right)\mathbf{1}^\top.
\end{aligned}$$

One step of proximal gradient descent is given as follows (details are given in Appendix B):

$$\mathbf{S}^{(k+1)} = \mathrm{prox}_{\eta_2\left(\mu_1\|\cdot\|_1 + \mathbb{I}_{\mathcal{S}}(\cdot)\right)}\left(\mathbf{S}^{(k)} - \eta_2\mathbf{T}^{(k)}\right), \tag{7}$$

where $\eta_2$ is the step size and $\mathbb{I}_{\mathcal{S}}(\mathbf{S})$ denotes the indicator function taking value 0 if $\mathbf{S} \in \mathcal{S}$ and $+\infty$ otherwise. Moreover, the proximal operator in Eq. (7) can be computed analytically as

$$\mathbf{S}^{(k+1)} = \min\Big\{1, \mathrm{ReLU}\big(\mathbf{S}^{(k)} - \eta_2 \mathbf{T}^{(k)} - \eta_2 \mu_1 \mathbf{1}\mathbf{1}^\top\big)\Big\},$$

where $\mathrm{ReLU}(\mathbf{X}) = \max\{\mathbf{0}, \mathbf{X}\}$. In conclusion, the overall procedure of ASMP can be summarized as follows:

$$\begin{cases} \mathbf{H}^{(k+1)} = (1 - 2\eta_1 - 2\eta_1\lambda)\,\mathbf{H}^{(k)} + 2\eta_1\lambda\mathrm{Diag}\big(\mathbf{S}^{(k)}\mathbf{1}\big)^{-1}\mathbf{S}^{(k)}\mathbf{H}^{(k)} + 2\eta_1\mathbf{X}, \\ \mathbf{S}^{(k+1)} = \min\big\{1, \mathrm{ReLU}\big(\mathbf{S}^{(k)} - \eta_2\mathbf{T}^{(k)} - \eta_2\mu_1\mathbf{1}\mathbf{1}^\top\big)\big\}, \end{cases} \quad k = 0, \ldots, K-1.$$

$$\text{(ASMP)}$$

The ASMP can be interpreted as the standard message passing (i.e., the update step of $\mathbf{H}$) with extra operations that adaptively adjust the graph structure (i.e., the update step of $\mathbf{S}$). Therefore, an edge of the graph included in some layers may be excluded or down-weighted in other layers. A pictorial illustration of the ASMP procedure is provided in Figure 1. A $K$-layer ASMP can be fully specified by parameters $\gamma, \lambda, \mu_1, \mu_2, \eta_1$, and $\eta_2$, which we generally denote as $\mathrm{ASMP}_K\big(\mathbf{X}, \mathbf{A}, \gamma, \lambda, \mu_1, \mu_2, \eta_1, \eta_2\big)$.

Note that ASMP is general enough to cover several existing propagation rules as special cases.

*Remark* 1 (Special cases). If we use a fixed graph structure $\mathbf{S}^{(0)} = \cdots = \mathbf{S}^{(K)} = \mathbf{A}$ in ASMP, i.e., $\mu_1 = \mu_2 = \gamma = 0$, the ASMP reduces to a classical message passing procedure that only performs feature learning. Specifically, with $\eta_1 = \frac{1}{2+2\lambda}$ and the symmetric normalized adjacency matrix, ASMP can be written as

$$\mathbf{H}^{(k+1)} = \frac{\lambda}{1+\lambda}\mathbf{A}_{\mathrm{sym}}\mathbf{H}^{(k)} + \frac{1}{1+\lambda}\mathbf{X}. \tag{8}$$

Case I: when $\lambda = \frac{1}{\alpha} - 1$, the operation in Eq. (8) becomes the message passing rule of APPNP (Klicpera et al., 2019):

$$\mathbf{H}^{(k+1)} = (1-\alpha)\,\mathbf{A}_{\mathrm{sym}}\mathbf{H}^{(k)} + \alpha\mathbf{X}.$$

Case II: when $\lambda = \infty$, the operation in Eq. (8) becomes the simple aggregation in many GNN models such as the GCN model (Kipf and Welling, 2017) and the simple graph convolution (SGC) model (Wu et al., 2019a):

$$\mathbf{H}^{(k+1)} = \mathbf{A}_{\mathrm{sym}}\mathbf{H}^{(k)}.$$

Instead of updating both $\mathbf{S}$ and $\mathbf{H}$ once, we can also choose to update them for several steps. The convergence of ASMP is guaranteed with proper selections of the step sizes as demonstrated in Theorem 4. Before proceeding to the convergence result, we first introduce some standard assumptions on the node feature vectors and the degree matrices, which are widely adopted in the literature (Garg et al., 2020; Liao et al., 2021; Cong et al., 2021).

**Assumption 2.** *The energy of the node feature is uniformly upperbounded, i.e., $\|\mathbf{H}_{i,:}^{(k)}\|_2 \leq B$ for $i = 1, \ldots, N$ and $k = 0, \ldots, K$.*

**Assumption 3.** *The diagonal elements of the degree matrix is lowerbounded by a positive constant, i.e., $\min_i \mathbf{D}_{ii} = c > 0$ for $i = 1, \ldots, N$.*

**Theorem 4.** *Let $\mathbf{H}^{(0)} = \mathbf{X}$ and $\mathbf{S}^{(0)} = \mathbf{A}$. Under Assumption 2 and Assumption 3, the sequence $\{\mathbf{H}^{(k)}, \mathbf{S}^{(k)}\}_{k=1}^K$ generated by (ASMP) with $0 < \eta_1 < \frac{1}{1+2\lambda}$ and $0 < \eta_2 < \frac{1}{\gamma + \mu_2 + (1 + \frac{1}{c}N\sqrt{N})\frac{\lambda}{c^2}N^2B^2}$ converges to a first-order stationary point of Problem (4) denoted as $\{\mathbf{H}^*, \mathbf{S}^*\}$ with rate*

$$\inf_{k \geq K}\Big\{\big\|\mathbf{H}^{(k+1)} - \mathbf{H}^{(k)}\big\|_{\mathrm{F}}^2 + \big\|\mathbf{S}^{(k+1)} - \mathbf{S}^{(k)}\big\|_{\mathrm{F}}^2\Big\} \leq \frac{1}{\rho K}\Big(p\big(\mathbf{H}^{(0)}, \mathbf{S}^{(0)}\big) - p\big(\mathbf{H}^*, \mathbf{S}^*\big)\Big),$$

*where $\rho$ is a constant depending on the step sizes and Lipschitz constants, and $p(\mathbf{H}, \mathbf{S})$ represents the objective of Problem (4).*

*Proof.* The proof for Theorem 4 is in Appendix C. Note that if multiple updating steps are used for $\mathbf{S}$ and $\mathbf{H}$ in ASMP, this convergence result still holds (Bolte et al., 2014; Nikolova and Tan, 2017). $\qquad\square$

### 3.3 ASGNN: Graph Neural Networks with Adaptive Structure

In this section, we introduce a family of GNNs leveraging the ASMP scheme. Integrating (ASMP) with a machine learning model $g_\theta(\cdot)$ (e.g., a multilayer perceptron) with $\mathbf{H}^{(0)} = \mathbf{X} = g_\theta(\mathbf{Z})$, a $K$-layer ASGNN model is defined as follows:

$$\mathbf{H}^{(K)} = \text{ASMP}_K\Big(\mathbf{H}^{(0)}, \mathbf{A}, \gamma, \lambda, \mu_1, \mu_2, \eta_1, \eta_2\Big).$$

In ASGNN, we have chosen a decoupled architecture similar to APPNP (Klicpera et al., 2019) and deep adaptive GNN (DAGNN) (Liu et al., 2021a). Specially, in ASGNN, the model $g_\theta$ will first transform the initial node feature as $\mathbf{X} = g_\theta(\mathbf{Z})$, and then ASMP performs $K$ steps of message passing with input $g_\theta(\mathbf{Z})$.

The parameters in ASMP, i.e., $\gamma$, $\lambda$, $\mu_1$, and $\mu_2$, are set to be weights to be learned from the downstream tasks. For example, in semi-supervised node classification tasks, the loss function is chosen as the cross-entropy classification loss on the labeled nodes and the whole model is trained in an end-to-end way. Since ASMP is derived from the alternating (proximal) gradient descent algorithm, a trained ASMP is naturally a parameter-optimized iterative algorithm. The step sizes $\eta_1$ and $\eta_2$ in ASMP can be chosen according to the results in Theorem 4. However, such choices seem to be too conservative in practice and may lead to slow convergence. Thus, we may also consider the step sizes $\eta_1$ and $\eta_2$ as learnable parameters. Convergence property of ASMP with learned step sizes will be showcased in the experiments. In conclusion, there are in total six parameters in ASMP considered during the learning process.

In this paper, we have focused on problems in which there is an initial graph structure, while the use of ASGNN may also be extended to scenarios where the initial structure is not available. In such case, we can first create a $k$-nearest neighbor graph or use some optimization methods (Dong et al., 2016; Kalofolias, 2016; Kumar et al., 2020) to learn a graph structure based on the node feature. Such extensions of ASGNN can be promising future research directions.

## 4 EXPERIMENTS

In this section, we conduct experiments to validate the effectiveness of the proposed ASGNN model. First, we introduce the experimental settings. Then, we assess the performance of ASGNN on semi-supervised node classifications tasks and investigate the benefits of introducing adaptive structure into GNNs against global attacks and targeted attacks. Finally, we analyze the structure denoising ability and the convergence property of ASMP with the learned step sizes.

### 4.1 EXPERIMENT SETTINGS

**Datasets**: We perform numerical experiments on 4 real-world citation graphs, i.e., Cora (Sen et al., 2008), Citeseer (Sen et al., 2008), Cora-ML (Bojchevski and Günnemann, 2018), and ACM (Wang et al., 2019), and only consider the largest connected component in each dataset.

**Baselines**: To evaluate the effectiveness of ASGNN, we compare it with GCN and several benchmarks that are designed from different perspectives to robustify the GNNs, including GCN-Jaccard (Wu et al., 2019b) that pre-processes the graph by eliminating edges with low Jaccard similarity of node feature vectors, GCN-SVD (Entezari et al., 2020) that applies the low-rank approximation of the given graph adjacency matrix, GNNGuard (Zhang and Zitnik, 2020) that defines an attention mechanism based on the similarity of neighboring nodes, Pro-GNN (Jin et al., 2020) that jointly learns a graph structure and a GNN model guided by some predefined structural priors, and Elastic GNN (Liu et al., 2021b) that utilizes trend filtering instead of Laplacian smoothing to promote robustness. The code is implemented based on PyTorch Geometric (Fey and Lenssen, 2019). For GCN-Jaccard, GCN-SVD, and Pro-GNN, we use the implementation provided in DeepRobust (Li et al., 2020). For GNNGuard and Elastic GNN, we follow the implementation provided in the original papers (Zhang and Zitnik, 2020; Liu et al., 2021b).

**Parameter settings**: For all the experimental results, we give the average performance and standard variance with 10 independent trials. For each graph, we randomly select 10%/10%/80% of nodes for training, validation, and testing. The Adam optimizer is used in all experiments. The models' hyperparameters are tuned based on the results of the validation set. The search space of hyperparameters

Table 1: Node classification performance (accuracy $\pm$ std) under global attack (**Bold**: the best model; wavy: the runner-up model)

| Dataset | Ptb. rate (%) | GCN | GCN-Jaccard | GCN-SVD | GNNGuard | Pro-GNN | Elastic GNN | ASGNN |
|---|---|---|---|---|---|---|---|---|
| Cora | 0 | 85.34 ± 0.39 | 81.75 ± 0.49 | 75.15 ± 0.64 | 80.14 ± 0.15 | 82.94 ± 0.28 | 84.80 ± 0.58 | **85.38 ± 0.24** |
| | 5 | 79.71 ± 0.48 | 77.81 ± 0.52 | 73.71 ± 0.42 | 78.71 ± 0.33 | 82.20 ± 0.35 | 82.26 ± 0.69 | **82.31 ± 0.53** |
| | 10 | 74.28 ± 0.79 | 74.38 ± 0.30 | 65.85 ± 0.39 | 75.35 ± 0.24 | 79.30 ± 0.64 | 79.47 ± 1.52 | **80.31 ± 0.61** |
| | 15 | 69.05 ± 0.77 | 72.54 ± 0.31 | 65.33 ± 0.47 | 73.54 ± 0.57 | 77.69 ± 0.74 | 77.84 ± 1.08 | **78.11 ± 0.76** |
| | 20 | 57.76 ± 1.01 | 71.76 ± 0.48 | 60.85 ± 0.74 | 70.99 ± 0.29 | 74.16 ± 1.02 | 63.68 ± 0.27 | **77.04 ± 0.59** |
| | 25 | 52.67 ± 1.00 | 69.67 ± 0.46 | 59.31 ± 0.47 | 68.79 ± 0.82 | 71.19 ± 1.27 | 62.90 ± 3.37 | **75.18 ± 0.97** |
| Citeseer | 0 | 73.97 ± 0.54 | 72.09 ± 0.49 | 68.34 ± 0.39 | 73.00 ± 0.50 | 73.35 ± 0.47 | 73.82 ± 0.43 | **73.99 ± 0.93** |
| | 5 | 72.57 ± 0.93 | 70.79 ± 0.30 | 67.59 ± 0.43 | 72.29 ± 0.39 | 73.16 ± 0.42 | 73.30 ± 0.37 | **73.35 ± 0.41** |
| | 10 | 71.21 ± 1.44 | 70.27 ± 0.62 | 67.38 ± 0.65 | 70.15 ± 0.36 | 72.78 ± 0.79 | 72.78 ± 0.66 | **72.83 ± 0.56** |
| | 15 | 68.00 ± 1.04 | 69.97 ± 1.49 | 66.47 ± 0.51 | 70.42 ± 0.57 | 71.55 ± 0.73 | 71.73 ± 1.03 | **71.85 ± 1.83** |
| | 20 | 59.75 ± 0.83 | 69.49 ± 0.71 | 65.83 ± 0.69 | 69.28 ± 0.41 | 70.07 ± 1.12 | 61.55 ± 1.82 | **71.06 ± 3.09** |
| | 25 | 59.98 ± 0.98 | 68.14 ± 0.36 | 62.34 ± 0.61 | 68.58 ± 0.52 | 69.73 ± 0.93 | 63.98 ± 2.17 | **70.03 ± 3.45** |
| Cora-ML | 0 | 86.59 ± 0.07 | 84.68 ± 0.32 | 82.96 ± 0.27 | 79.25 ± 0.30 | 79.48 ± 0.40 | **87.01 ± 0.28** | 86.68 ± 0.43 |
| | 5 | 80.99 ± 0.50 | 81.80 ± 0.37 | 81.78 ± 0.46 | 79.07 ± 0.11 | 78.57 ± 0.16 | 84.68 ± 0.25 | **84.80 ± 0.80** |
| | 10 | 74.57 ± 0.75 | 80.35 ± 0.24 | 81.75 ± 0.33 | 78.99 ± 0.10 | 78.74 ± 0.84 | 82.01 ± 0.64 | **83.09 ± 0.59** |
| | 15 | 54.69 ± 0.52 | 76.53 ± 0.29 | 74.76 ± 0.44 | **78.59 ± 0.13** | 73.62 ± 0.85 | 64.59 ± 2.69 | 73.71 ± 1.82 |
| | 20 | 40.24 ± 1.97 | 76.46 ± 0.58 | 53.94 ± 0.45 | **77.58 ± 0.13** | 72.72 ± 0.88 | 52.18 ± 0.71 | 73.65 ± 1.42 |
| | 25 | 44.13 ± 3.42 | 75.95 ± 0.50 | 71.98 ± 0.17 | **77.03 ± 0.11** | 74.91 ± 0.56 | 53.05 ± 0.36 | 75.36 ± 1.34 |
| ACM | 0 | 91.75 ± 0.10 | 89.62 ± 0.41 | 87.51 ± 0.42 | 91.60 ± 0.29 | 90.11 ± 0.57 | 91.45 ± 0.21 | **92.56 ± 0.42** |
| | 5 | 84.29 ± 0.57 | 84.64 ± 0.27 | 85.29 ± 1.13 | 82.64 ± 1.34 | 88.25 ± 1.19 | 90.10 ± 0.27 | **90.60 ± 0.28** |
| | 10 | 81.71 ± 0.61 | 81.12 ± 0.31 | 84.59 ± 0.68 | 80.32 ± 1.29 | 88.14 ± 0.60 | 89.45 ± 0.41 | **90.10 ± 0.35** |
| | 15 | 79.65 ± 1.00 | 74.66 ± 0.94 | 83.81 ± 0.81 | 77.44 ± 1.46 | 87.59 ± 0.74 | 89.23 ± 0.34 | **89.93 ± 0.51** |
| | 20 | 79.95 ± 0.50 | 74.26 ± 0.75 | 82.35 ± 1.64 | 77.38 ± 1.55 | 87.83 ± 1.03 | 88.65 ± 0.35 | **90.61 ± 0.28** |
| | 25 | 79.55 ± 1.16 | 74.12 ± 0.81 | 82.04 ± 0.99 | 77.60 ± 1.60 | 88.06 ± 0.85 | 88.15 ± 0.58 | **90.15 ± 0.33** |

are as follows: 1) learning rate: {0.005, 0.01, 0.05}; 2) weight decay: {0, 5e-5, 5e-4}; 3) dropout rate: {0.1, 0.5, 0.8}; 4) model depth: {2, 4, 8, 16}. For GCN-Jaccard, the threshold of Jaccard similarity for removing dissimilar edges is chosen from {0.01, 0.02, 0.03, 0.04, 0.05, 0.1}. For GCN-SVD, the reduced rank of the graph is tuned from {5, 10, 15, 50, 100, 200}. For Elastic GNN, the regularization coefficients are chosen from {3, 6, 9}. For Pro-GNN, we adopt the hyperparameters provided in their paper (Jin et al., 2020).

## 4.2 PERFORMANCE UNDER ADVERSARIAL ATTACK

The performance of the compared models is evaluated under the training-time adversarial attacks (Wang and Gong, 2019; Zügner and Günnemann, 2019), i.e., the graph is first attacked, and then the GNN models are trained on the perturbed graph. In the following, we conduct experiments under both the global attack and the targeted attack. Specifically, the global attack aims to reduce the overall performance of GNNs (Zügner and Günnemann, 2019) while the targeted attack aims to fool GNNs on some specific nodes (Zügner et al., 2018).

**Global Attack**: We first test the node classification performance of ASGNN and other baselines under global attack using a representative global attack method called meta-attack (Zügner and Günnemann, 2019). We vary the perturbation rate, i.e., the ratio of changed edges, from 0% to 25% with an increasing step of 5%. The results are reported in Table 1. From the table, we observe that the proposed ASGNN model outperforms other methods in most cases. For instance, ASGNN improves GCN over 30% on the Cora-ML dataset at a 20% perturbation rate and over 20% on the Cora dataset at a 25% perturbation rate. On Cora, Citeseer, and ACM datasets, ASGNN beats other baselines at various perturbation rates by a large margin. The GCN-Jaccard method and the GNNGuard slightly outperforms ASGNN on the Cora-ML dataset at a 15%-25% perturbation rate, while they performs poorly on other datasets. Specifically, on the other three datasets under the 25% perturbation rate, ASGNN outperforms GCN-Jaccard by 22%, 10%, and 10%, respectively. Such inspiring results demonstrate that ASGNN can better resist global attack than other baseline methods.

**Targeted Attack**: For the targeted attack, we use a representative method called NETTACK (Zügner et al., 2018). Following existing works (Zhu et al., 2019; Jin et al., 2020), we vary the perturbation number made on every node, i.e., the number of edge removals/additions, from 0 to 5 with an increasing step of 1. The results are reported in Table 2. We choose the nodes in the test set with degrees larger than 10 as targeted nodes and the reported classification performance is evaluated on target nodes. Thus, the results in Table 2 is not directly comparable with the results in Table 1. From the table, we can see that the proposed ASGNN attains better performance than other baselines in most cases. For instance, on the Citeseer dataset with 5 perturbations per targeted node, ASGNN

Table 2: Node classification performance (accuracy ± std) under targeted attack (**Bold**: the best model; wavy: the runner-up model)

| Dataset | Ptb. number | GCN | GCN-Jaccard | GCN-SVD | GNNGuard | Pro-GNN | Elastic GNN | ASGNN |
|---|---|---|---|---|---|---|---|---|
| Cora | 0 | 82.53 ± 1.45 | 81.95 ± 0.29 | 77.35 ± 1.40 | 77.59 ± 1.34 | 82.92 ± 0.29 | **84.93 ± 2.28** | 83.01 ± 1.57 |
| | 1 | 78.19 ± 1.66 | 75.30 ± 1.54 | 75.18 ± 1.80 | 76.67 ± 2.93 | 81.48 ± 0.91 | 81.44 ± 1.81 | **81.57 ± 1.18** |
| | 2 | 71.33 ± 1.29 | 70.24 ± 1.52 | 71.81 ± 1.63 | 75.48 ± 3.02 | **79.03 ± 1.80** | 76.74 ± 1.97 | 78.80 ± 1.03 |
| | 3 | 66.63 ± 1.53 | 69.04 ± 0.94 | 65.18 ± 1.65 | 75.09 ± 1.06 | 72.75 ± 1.32 | 73.97 ± 2.67 | **75.30 ± 1.35** |
| | 4 | 61.45 ± 2.16 | 61.68 ± 1.05 | 58.79 ± 2.14 | **70.45 ± 1.34** | 70.11 ± 2.45 | 68.31 ± 3.50 | 70.24 ± 4.70 |
| | 5 | 56.75 ± 1.37 | 59.52 ± 1.88 | 59.16 ± 2.71 | **70.58 ± 1.80** | 66.98 ± 1.63 | 65.78 ± 2.51 | 68.55 ± 3.21 |
| Citeseer | 0 | 81.27 ± 0.95 | 80.31 ± 1.26 | 80.47 ± 1.01 | **83.65 ± 1.75** | 81.24 ± 1.01 | 81.42 ± 0.76 | 81.90 ± 1.95 |
| | 1 | 80.63 ± 0.63 | 80.00 ± 1.45 | 78.57 ± 2.67 | 80.95 ± 2.75 | 80.52 ± 0.85 | 80.79 ± 1.17 | **81.21 ± 1.11** |
| | 2 | 79.84 ± 1.02 | 76.98 ± 1.77 | 73.02 ± 6.77 | 78.36 ± 5.53 | 80.63 ± 0.95 | 81.01 ± 0.50 | **81.11 ± 1.32** |
| | 3 | 66.51 ± 3.36 | 74.76 ± 1.31 | 76.03 ± 3.71 | 78.09 ± 3.39 | 79.36 ± 4.76 | 80.31 ± 1.10 | **80.32 ± 1.90** |
| | 4 | 62.54 ± 1.62 | 76.34 ± 1.49 | 62.22 ± 3.31 | 77.69 ± 4.63 | 75.71 ± 4.87 | 72.06 ± 5.60 | **80.16 ± 1.28** |
| | 5 | 52.70 ± 1.98 | 72.85 ± 1.65 | 60.16 ± 6.67 | 75.37 ± 3.15 | 73.95 ± 7.13 | 73.96 ± 3.90 | **80.75 ± 1.70** |
| Cora-ML | 0 | 88.33 ± 0.56 | 84.91 ± 0.24 | 83.06 ± 0.30 | 81.82 ± 0.45 | 86.64 ± 0.80 | 88.84 ± 0.67 | **88.88 ± 0.31** |
| | 1 | 83.85 ± 0.47 | 83.82 ± 0.33 | 80.51 ± 0.28 | 82.22 ± 0.66 | 83.77 ± 0.96 | 85.87 ± 1.01 | **86.20 ± 1.21** |
| | 2 | 79.18 ± 1.31 | 83.75 ± 0.33 | 78.73 ± 0.34 | 82.35 ± 0.66 | 82.29 ± 0.13 | 84.34 ± 1.02 | **84.41 ± 1.51** |
| | 3 | 76.19 ± 0.82 | 82.18 ± 0.48 | 78.23 ± 0.20 | 82.39 ± 0.33 | 80.58 ± 1.06 | 81.41 ± 1.62 | **83.50 ± 1.08** |
| | 4 | 70.61 ± 1.56 | **81.90 ± 0.45** | 76.25 ± 0.51 | 80.50 ± 0.61 | 79.92 ± 1.44 | 78.31 ± 1.20 | 80.81 ± 0.99 |
| | 5 | 64.52 ± 1.29 | **81.35 ± 0.34** | 75.47 ± 0.41 | 79.86 ± 0.54 | 77.63 ± 1.68 | 74.08 ± 1.89 | 80.17 ± 3.85 |
| ACM | 0 | 90.33 ± 0.09 | 89.76 ± 0.39 | 86.21 ± 0.46 | 90.36 ± 0.19 | 90.22 ± 0.60 | 90.31 ± 0.71 | **92.11 ± 0.41** |
| | 1 | 89.70 ± 0.22 | 83.62 ± 0.47 | 83.50 ± 0.71 | 89.09 ± 0.35 | 87.09 ± 0.65 | 90.03 ± 0.43 | **91.08 ± 1.36** |
| | 2 | 82.06 ± 1.12 | 80.47 ± 0.63 | 82.09 ± 0.73 | 81.86 ± 1.99 | 87.01 ± 0.53 | 87.72 ± 1.27 | **90.55 ± 0.47** |
| | 3 | 80.26 ± 1.17 | 77.07 ± 0.67 | 81.09 ± 0.78 | 79.24 ± 1.91 | 87.07 ± 1.14 | 84.75 ± 2.07 | **89.54 ± 0.49** |
| | 4 | 76.86 ± 1.46 | 77.45 ± 0.44 | 80.76 ± 0.54 | 74.88 ± 2.04 | 87.04 ± 1.16 | 83.49 ± 2.01 | **89.45 ± 0.51** |
| | 5 | 73.32 ± 1.77 | 74.38 ± 0.59 | 79.74 ± 0.77 | 71.54 ± 1.97 | 86.42 ± 0.71 | 81.67 ± 1.53 | **88.44 ± 0.71** |

improves GCN by 25% and outperforms other baselines by over 5%. The reported results demonstrate that ASGNN can also effectively resist the targeted attack.

### 4.3 ROBUSTNESS OF ASGNN

The message passing scheme in ASGNN is designed based on jointly node feature learning and graph structure learning principle. To validate that ASGNN can help purify (i.e., denoise) the structure, we evaluate the quality of the graphs generated by ASGNN via evaluating the performance of a GCN model trained on the generated graphs in the last layers of ASGNN (which we named as GCN-AS). Under the assumption that the GCN trained on a purer graph will give better performance, the performance of GCN-AS can indicate the quality of the learned graph in ASGNN. We conduct experiments on different datasets under targeted attack with five perturbations per node and the results are in Table 3. It can be seen that the performance of GCN-AS is much better than GCN. Moreover, GCN-AS even outperforms other defense models on some datasets, e.g., GCN-AS outperforms all other baselines on ACM dataset. These inspiring results indicate that ASGNN can mitigate the influence of adversarial attacks.

Table 3: Performance comparison of GCN and GCN-AS.

| Model | Cora | Citeseer | Cora-ML | ACM |
|---|---|---|---|---|
| GCN | 56.75 ± 1.37 | 52.70 ± 1.98 | 64.52 ± 1.29 | 73.32 ± 1.77 |
| GCN-AS | 67.91 ± 9.28 | 69.96 ± 5.30 | 79.63 ± 7.98 | 87.56 ± 1.49 |

We provide some additional experiments in Appendix E, including the runtime analysis in Appendix E.1, the discussion of the leaned coeefficients in ASGNN in Appendix E.2, the sparsity level of the graphs generated in ASGNN in Appendix E.3, and the convergence analysis of ASMP with learned step sizes in Appendix E.4.

## 5 CONCLUSION

In this work, we have developed an interpretable robust message passing scheme named ASMP following the jointly node feature learning and graph structure learning principle. ASMP is provably convergent and it has a clear interpretation as a standard message passing scheme with adaptive structure. Integrating ASMP with neural network components, we have obtained a family of robust graph neural networks with adaptive structure. Extensive experiments on real-world datasets with various adversarial attack settings corroborate the effectiveness and the robustness of the proposed graph neural network architecture.

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

## A    RELATED WORK ON OPTIMIZATION-INDUCED GRAPH NEURAL NETWORK DESIGN

Since ASGNN proposed in this paper is induced from an optimization algorithm, in this section, we give more literatue review on optimization-induced GNN model design to supplement our discussion.

The idea of optimization-induced GNN model design partly stems from the observation that many primitive handcrafted GNN models could be nicely interpreted as (unrolled) iterative algorithms for solving a GSD optimization problem (Ma et al., 2021; Zhu et al., 2021; Zhang and Zhao, 2022). Based on this observation, many papers aim at strengthening the capability of GNNs by carefully designing the underlying optimization problems and/or the iterative algorithms solving it.

For example, inspired by the idea of trend filtering (Wang et al., 2015), Liu et al. (2021b) replace the Laplacian smoothing term (which is in the form of $\ell_2$-norm) in the GSD problem with an $\ell_{2,1}$-norm to promote robustness against abnormal edges. Also for robustness pursuit, Yang et al. (2021) replace the Laplacian smoothing term with nonlinear functions imposed over pairwise node distances. Since the classical Laplacian smoothing term in GSD only promotes smoothness over connected nodes, the authors in Zhang et al. (2020); Zhao and Akoglu (2020) further suggest promoting the non-smoothness over the disconnected nodes, which is achieved by deducting the sum of distances between disconnected pairs of nodes from the denoising objective. In Jiang et al. (2022), the authors augment the GSD objective with a fairness term to fight against large topology bias. Most recently, Fu et al. (2022) propose a $p$-Laplacian message passing scheme and a ${}^p$GNN model, which is capable of dealing with heterophilic graphs and is robust to adversarial perturbations. Apart from that, Ahn et al. (2022) designs a novel regularization term to build heterogeneous GNNs.

Although there is rich literature on optimization-induced GNN model design, all of them are focusing on learning the node feature matrix. The idea of this paper is similar to them in terms of the GNN design philosophy, however, we design an objective to jointly learn the node feature and the graph structure which was rarely covered in the literature.

## B    DERIVATION OF THE PROXIMAL GRADIENT STEP IN EQ. (7)

For the $\mathbf{S}$-block optimization, i.e., Eq. (6), we define the objective function except the term $\mu_1 \|\mathbf{S}\|_1$ as $f_S(\mathbf{S})$, i.e.,

$$f_S(\mathbf{S}) = \gamma\|\mathbf{S} - \mathbf{A}\|_{\mathrm{F}}^2 - \lambda\mathrm{Tr}\left(\mathbf{H}^\top \mathbf{D}^{-1}\mathbf{S}\mathbf{H}\right) + \mu_2 \|\mathbf{S}\|_{\mathrm{F}}^2 , \tag{9}$$

where $\mathbf{D} = \mathrm{Diag}\left(\mathbf{S}\mathbf{1}\right)$. In this section, we first derive the expression of $\nabla f_S(\mathbf{S})$ and then compute the proximal operator in Eq. (7).

### B.1    ON COMPUTATION OF $\nabla f_S(\mathbf{S})$

We first focus on the gradient computation of the second term in Eq. (9). For the graph degree matrix, we have

$$\mathbf{D} = \mathrm{Diag}\left(\mathbf{S}\mathbf{1}\right) = \mathbf{S}\mathbf{1}\mathbf{1}^\top \odot \mathbf{I},$$

where $\odot$ denotes the Hadamard product. Based on the rule of matrix calculus, the differential of the scalar function $\mathrm{Tr}(\mathbf{H}^\top \mathbf{D}^{-1}\mathbf{S}\mathbf{H})$ with respect to matrix variable $\mathbf{S}$ can be computed as follows:

$$\mathrm{d}\left(\mathrm{Tr}\left(\mathbf{H}^\top \mathbf{D}^{-1}\mathbf{S}\mathbf{H}\right)\right) = \mathrm{Tr}\left(\mathbf{H}^\top \mathbf{D}^{-1}\mathrm{d}\left(\mathbf{S}\right)\mathbf{H} + \mathbf{H}^\top \mathrm{d}\left(\mathbf{D}^{-1}\right)\mathbf{S}\mathbf{H}\right).$$

For an invertible $\mathbf{D}$ (note that, in this paper, the graphs considered all have self loops, so $\mathbf{D}$ is always invertible), we have

$$\mathrm{d}\left(\mathbf{D}^{-1}\right) = -\mathbf{D}^{-1}\mathrm{d}\left(\mathbf{D}\right)\mathbf{D}^{-1}.$$

Thus, we can get

$$\begin{aligned}
\mathrm{d}\left(\mathrm{Tr}\left(\mathbf{H}^\top \mathbf{D}^{-1}\mathbf{S}\mathbf{H}\right)\right) =& \mathrm{Tr}\left(\mathbf{H}^\top \mathbf{D}^{-1}\mathrm{d}\left(\mathbf{S}\right)\mathbf{H} - \mathbf{H}^\top \mathbf{D}^{-1}\mathrm{d}(\mathbf{S}\mathbf{1}\mathbf{1}^\top \odot \mathbf{I})\mathbf{D}^{-1}\mathbf{S}\mathbf{H}\right) \\
=& \mathrm{Tr}\left(\mathbf{H}\mathbf{H}^\top \mathbf{D}^{-1}\mathrm{d}\left(\mathbf{S}\right) - \left(\mathbf{D}^{-1}\mathbf{S}\mathbf{H}\mathbf{H}^\top \mathbf{D}^{-1} \odot \mathbf{I}\right)\mathrm{d}\left(\mathbf{S}\right)\mathbf{1}\mathbf{1}^\top\right) \\
=& \mathrm{Tr}\left(\left(\mathbf{H}\mathbf{H}^\top \mathbf{D}^{-1} - \mathbf{1}\mathrm{Diag}\left(\mathbf{D}^{-1}\mathbf{S}\mathbf{H}\mathbf{H}^\top \mathbf{D}^{-1}\right)^\top\right)\mathrm{d}\left(\mathbf{S}\right)\right).
\end{aligned}$$

Since $\mathrm{d}\left(\mathrm{Tr}(\mathbf{H}^\top \mathbf{D}^{-1}\mathbf{S}\mathbf{H})\right) = \mathrm{Tr}\left(\left(\frac{\mathrm{dTr}(\mathbf{H}^\top \mathbf{D}^{-1}\mathbf{S}\mathbf{H})}{\mathrm{d}\mathbf{S}}\right)^\top \mathrm{d}(\mathbf{S})\right)$, we have

$$\frac{\mathrm{dTr}\left(\mathbf{H}^\top \mathbf{D}^{-1}\mathbf{S}\mathbf{H}\right)}{\mathrm{d}\mathbf{S}} = \mathbf{D}^{-1}\mathbf{H}\mathbf{H}^\top - \mathrm{Diag}\left(\mathbf{D}^{-1}\mathbf{S}\mathbf{H}\mathbf{H}^\top \mathbf{D}^{-1}\right)\mathbf{1}^\top.$$

For other terms in $f_S(\mathbf{S})$, the gradients with respect to $\mathbf{S}$ can be easily computed. Finally, we obtain

$$\nabla f_S(\mathbf{S}) = 2\gamma\left(\mathbf{S} - \mathbf{A}\right) - \lambda\left(\mathbf{D}^{-1}\mathbf{H}\mathbf{H}^\top - \mathrm{Diag}\left(\mathbf{D}^{-1}\mathbf{S}\mathbf{H}\mathbf{H}^\top \mathbf{D}^{-1}\right)\mathbf{1}^\top\right) + 2\mu_2\mathbf{S}.$$

## B.2 On Computation of The Proximal Step

**Lemma 5.** *Given a matrix* $\mathbf{M} \in \mathbb{R}^{N \times N}$, *we have*

$$\mathrm{prox}_{\kappa\|\cdot\|_1 + \mathbb{I}_S(\cdot)}\left(\mathbf{M}\right) = \min\left\{1, \mathrm{ReLU}\left(\mathbf{M} - \kappa\mathbf{1}\mathbf{1}^\top\right)\right\}, \tag{10}$$

*where* $\mathrm{ReLU}(\mathbf{X}) = \max\{\mathbf{0}, \mathbf{X}\}$.

*Proof.* The proximal step in Eq. (10) can be rewritten as the following optimization problem:

$$\underset{\mathbf{S} \in \mathcal{S}}{\text{minimize}} \quad \frac{1}{2}\|\mathbf{S} - \mathbf{M}\|_{\mathrm{F}}^2 + \kappa\|\mathbf{S}\|_1. \tag{11}$$

It is easy to observe that Problem (11) is decoupled over different elements in matrix $\mathbf{S}$. Therefore, each $\mathbf{S}_{ij}$ with $i, j = 1, \ldots, N$ can be optimized individually by solving the following optimization problem:

$$\underset{0 \leq \mathbf{S}_{ij} \leq 1}{\text{minimize}} \quad h_{ij}(\mathbf{S}_{ij}) = \frac{1}{2}\|\mathbf{S}_{ij} - \mathbf{M}_{ij}\|_{\mathrm{F}}^2 + \kappa|\mathbf{S}_{ij}|. \tag{12}$$

According to Eq. (10), we have

$$\mathbf{S}_{ij}^\star = \min\left\{1, \mathrm{ReLU}\left(\mathbf{M}_{ij} - \kappa\right)\right\}$$
$$= \begin{cases} 1 & 1 + \kappa \leq \mathbf{M}_{ij} \\ \mathbf{M}_{ij} - \kappa & \kappa \leq \mathbf{M}_{ij} < 1 + \kappa \\ 0 & \mathbf{M}_{ij} < \kappa. \end{cases}$$

Then, Lemma 5 can be proved by showing that for $i, j = 1, \ldots N$, $\mathbf{S}_{ij}^\star$ is the optimal solution for Problem (12). The optimality of $\mathbf{S}_{ij}^\star$ can be validated by verifying the optimality condition, i.e., there exists a subgradient $\psi \in \partial h_{ij}\left(\mathbf{S}_{ij}^\star\right)$ such that

$$\psi\left(\mathbf{S}_{ij} - \mathbf{S}_{ij}^\star\right) \geq 0$$

for all $0 \leq \mathbf{S}_{ij} \leq 1$. Observe that the subdifferential of $h_{ij}(\mathbf{S}_{ij})$ is computed as follows:

$$\partial h_{ij}(\mathbf{S}_{ij}) = \begin{cases} \mathbf{S}_{ij} - \mathbf{M}_{ij} + \kappa & \mathbf{S}_{ij} > 0 \\ \mathbf{S}_{ij} - \mathbf{M}_{ij} + \kappa\epsilon & \mathbf{S}_{ij} = 0, \end{cases}$$

where $\epsilon$ can be any constant satisfying $-1 \leq \epsilon \leq 1$. Then, the subdifferential $\partial h\left(\mathbf{S}_{ij}^\star\right)$ is given by

$$\partial h_{ij}(\mathbf{S}_{ij}^\star) = \begin{cases} 1 - \mathbf{M}_{ij} + \kappa & 1 + \kappa \leq \mathbf{M}_{ij} \\ 0 & \kappa \leq \mathbf{M}_{ij} < 1 + \kappa \\ -\mathbf{M}_{ij} + \kappa\epsilon & \mathbf{M}_{ij} < \kappa. \end{cases}$$

In the following, we will show that the optimality condition holds for each of the above cases.

1. For $1 + \kappa \leq \mathbf{M}_{ij}$, we have $\mathbf{S}_{ij}^\star = 1$ and $\psi = 1 - \mathbf{M}_{ij} + \kappa \leq 0$. Since $\mathbf{S}_{ij} \leq 1$, we can get $\psi(\mathbf{S}_{ij} - \mathbf{S}_{ij}^\star) \geq 0$.

2. For $\kappa \leq \mathbf{M}_{ij} < 1 + \kappa$, we have $\psi = 0$ and hence, $\psi(\mathbf{S}_{ij} - \mathbf{S}_{ij}^\star) = 0$ for all $0 \leq \mathbf{S}_{ij} \leq 1$.

3. For $\mathbf{M}_{ij} < \kappa$, we have $\mathbf{S}_{ij}^\star = 0$ and $\psi = -\mathbf{M}_{ij} + \kappa\epsilon$ with $\epsilon$ being any constant satisfying $-1 \leq \epsilon \leq 1$. Thus, we can choose $\epsilon = 1$, which leads to $\psi > 0$. Since $\mathbf{S}_{ij} \geq 0$, we can get $\psi(\mathbf{S}_{ij} - \mathbf{S}_{ij}^\star) \geq 0$.

In conclusion, there exists a subgradient $\psi \in \partial h_{ij}(\mathbf{S}_{ij}^\star)$ such that $\psi(\mathbf{S}_{ij} - \mathbf{S}_{ij}^\star) \geq 0$ for all $0 \leq \mathbf{S}_{ij} \leq 1$, based on which the optimality of Eq. (10) is validated and the proof is completed. $\qquad\square$

Based on the result in Lemma 5, by choosing $\mathbf{M} = \mathbf{S}^{(k)} - \eta_2 \nabla f_S(\mathbf{S}^{(k)})$ and $\kappa = \eta_2 \mu_1$, we get the analytical expression for the proximal step in Eq. (7) as follows:

$$\mathbf{S}^{(k+1)} = \min\left\{1, \mathrm{ReLU}\left(\mathbf{S}^{(k)} - \eta_2 \nabla f_S(\mathbf{S}^{(k)}) - \eta_2 \mu_1 \mathbf{1}\mathbf{1}^\top\right)\right\},$$

which suffices to perform the soft-thresholding operation and then project the solution onto the constraint set $\mathcal{S}$.

## C  PROOF OF THEOREM 4 (CONVERGENCE OF ASMP)

In this section, we will first prove that the objective function at the $\mathbf{H}$-block optimization problem and the smooth part of the objective function at the $\mathbf{S}$-block optimization problem are $L$-smooth. Then we give the conditions to ensure convergence of ASMP.

Denote $f_H(\mathbf{H})$ as the objective function at the $\mathbf{H}$-block optimization problem, i.e.,

$$f_H(\mathbf{H}) = \|\mathbf{H} - \mathbf{X}\|_F^2 + \lambda \mathrm{Tr}\left(\mathbf{H}^\top \mathbf{L}_{\mathrm{rw}} \mathbf{H}\right).$$

The $L$-smoothness of $f_H(\mathbf{H})$ is demonstrated in the following lemma.

**Lemma 6.** *Function $f_H(\mathbf{H})$ is $L$-smooth with $L_H = 2 + 4\lambda$, i.e., for any $\mathbf{H}_1, \mathbf{H}_2 \in \mathbb{R}^{N \times M}$, the following inequality holds:*

$$\|\nabla f_H(\mathbf{H}_1) - \nabla f_H(\mathbf{H}_2)\|_F \leq L_H \|\mathbf{H}_1 - \mathbf{H}_2\|_F.$$

*Proof.* First observe that

$$\begin{aligned}
&\|\nabla f_H(\mathbf{H}_1) - \nabla f_H(\mathbf{H}_2)\|_F \\
&= \left\|2(\mathbf{H}_1 - \mathbf{X}) + 2\lambda \mathbf{L}_{\mathrm{rw}}\mathbf{H}_1 - \left(2(\mathbf{H}_2 - \mathbf{X}) + 2\lambda \mathbf{L}_{\mathrm{rw}}\mathbf{H}_2\right)\right\|_F \\
&= 2\left\|(\mathbf{I} + \lambda \mathbf{L}_{\mathrm{rw}})(\mathbf{H}_1 - \mathbf{H}_2)\right\|_F \\
&\leq 2\left\|\mathbf{I} + \lambda \mathbf{L}_{\mathrm{rw}}\right\|_2 \|\mathbf{H}_1 - \mathbf{H}_2\|_F.
\end{aligned}$$

**Lemma 7** (Chung, 1997). *The largest eigenvalue of a random walk normalized Laplacian matrix $\mathbf{L}_{\mathrm{rw}}$ is less than or equal to 2, i.e., $\|\mathbf{L}_{\mathrm{rw}}\|_2 \leq 2$.*

Based on Lemma 7, we can conclude that

$$\|\nabla f_H(\mathbf{H}_1) - \nabla f_H(\mathbf{H}_2)\|_F \leq (2 + 4\lambda)\|\mathbf{H}_1 - \mathbf{H}_2\|_F.$$

Therefore, function $f_H(\mathbf{H})$ is $L$-smooth with $L_H = 2 + 4\lambda$ and the proof is completed $\qquad\square$

With $f_S$ defined in Eq. (9), the $L$-smoothness of $f_S$ is deomnstrated in the following lemma.

**Lemma 8.** *Function $f_S(\mathbf{S})$ is $L$-smooth with $L_S = 2\gamma + 2\mu_2 + \frac{2\lambda}{c^2}N^2 B^2 + \frac{2\lambda}{c^3}N^3\sqrt{N}B^2$, i.e., for any $\mathbf{S}_1, \mathbf{S}_2 \in \mathbb{R}^{N \times N}$, the following inequality holds:*

$$\|\nabla f_S(\mathbf{S}_1) - \nabla f_S(\mathbf{S}_2)\|_F \leq L_S \|\mathbf{S}_1 - \mathbf{S}_2\|_F.$$

*Proof.* Denote $\mathbf{D}_1 = \mathrm{Diag}(\mathbf{S}_1 \mathbf{1})$ and $\mathbf{D}_2 = \mathrm{Diag}(\mathbf{S}_2 \mathbf{1})$ as two degree matrices corresponding to $\mathbf{S}_1$ and $\mathbf{S}_2$. We have

$$\begin{aligned}
&\|\nabla f_S(\mathbf{S}_1) - \nabla f_S(\mathbf{S}_2)\|_F \\
&= \left\|2\gamma(\mathbf{S}_1 - \mathbf{A}) + 2\mu_2 \mathbf{S}_1 - \lambda \mathbf{D}_1^{-1}\mathbf{H}\mathbf{H}^\top + \lambda \mathrm{Diag}\left(\mathbf{D}_1^{-1}\mathbf{S}_1 \mathbf{H}\mathbf{H}^\top \mathbf{D}_1^{-1}\right)\mathbf{1}^\top \right. \\
&\qquad \left. - \left(2\gamma(\mathbf{S}_2 - \mathbf{A}) + 2\mu_2 \mathbf{S}_2 - \lambda \mathbf{D}_2^{-1}\mathbf{H}\mathbf{H}^\top + \lambda \mathrm{Diag}\left(\mathbf{D}_2^{-1}\mathbf{S}_2 \mathbf{H}\mathbf{H}^\top \mathbf{D}_2^{-1}\right)\mathbf{1}^\top\right)\right\|_F \\
&\leq (2\gamma + 2\mu_2)\|\mathbf{S}_1 - \mathbf{S}_2\|_F + \lambda \left\|\left(\mathbf{D}_1^{-1} - \mathbf{D}_2^{-1}\right)\mathbf{H}\mathbf{H}^\top\right\|_F \\
&\qquad + \lambda\left\|\mathrm{Diag}\left(\mathbf{D}_1^{-1}\mathbf{S}_1 \mathbf{H}\mathbf{H}^\top \mathbf{D}_1^{-1} - \mathbf{D}_2^{-1}\mathbf{S}_2 \mathbf{H}\mathbf{H}^\top \mathbf{D}_2^{-1}\right)\mathbf{1}^\top\right\|_F \\
&\leq (2\gamma + 2\mu_2)\|\mathbf{S}_1 - \mathbf{S}_2\|_F + \lambda\left\|\mathbf{H}\mathbf{H}^\top\right\|_2\left\|\mathbf{D}_1^{-1} - \mathbf{D}_2^{-1}\right\|_F \\
&\qquad + \lambda\sqrt{N}\left\|\mathbf{D}_1^{-1}\mathbf{S}_1 \mathbf{H}\mathbf{H}^\top \mathbf{D}_1^{-1} - \mathbf{D}_2^{-1}\mathbf{S}_2 \mathbf{H}\mathbf{H}^\top \mathbf{D}_2^{-1}\right\|_F.
\end{aligned} \qquad (13)$$

To derive the Lipschitz constant of $\nabla f_S(\mathbf{S})$, we first present several useful lemmas.

**Lemma 9.** *Under Assumption 2 that the norm of node feature vectors is upperbounded, i.e.,* $\left\| \mathbf{H}_{i,:} \right\|_2 \leq B$, *we have*

$$\left\| \mathbf{H}\mathbf{H}^\top \right\|_2 \leq \left\| \mathbf{H}\mathbf{H}^\top \right\|_F = \sqrt{\sum_{i=1}^{N} \sum_{j=1}^{N} \left( \mathbf{H}_{i,:}^\top \mathbf{H}_j \right)^2} \leq \sqrt{\sum_{i=1}^{N} \sum_{j=1}^{N} B^4} = NB^2.$$

**Lemma 10.** *Given* $\mathbf{S} \in \mathcal{S}$ *and* $\mathbf{D} = \mathrm{Diag}\,(\mathbf{S}\mathbf{1})$, *under Assumption 3 that the diagonal elements of* $\mathbf{D}$ *is lowerbounded by a positive constant, i.e.,* $\min_i \mathbf{D}_{ii} = c > 0$ *for* $i = 1, \ldots, N$, *we have*

$$\left\| \mathbf{D}^{-1}\mathbf{S} \right\|_2 \leq \left\| \mathbf{D}^{-1}\mathbf{S} \right\|_F = \sqrt{\sum_{i=1}^{N} \sum_{j=1}^{N} \left( \frac{\mathbf{S}_{ij}}{\mathbf{D}_{ii}} \right)^2} \leq \frac{N}{c}.$$

**Lemma 11.** *Given* $\mathbf{S}_1, \mathbf{S}_2 \in \mathcal{S}$, $\mathbf{D}_1 = \mathrm{Diag}\,(\mathbf{S}_1\mathbf{1})$, *and* $\mathbf{D}_2 = \mathrm{Diag}\,(\mathbf{S}_2\mathbf{1})$, *under Assumption 3 that the diagonal elements of the degree matrix is lowerbounded by a positive constant, i.e.,* $\min_i \mathbf{D}_{ii} = c > 0$ *for* $i = 1, \ldots, N$, *we have*

$$\left\| \mathbf{D}_1^{-1} - \mathbf{D}_2^{-1} \right\|_F \leq \frac{1}{c^2} N \left\| \mathbf{S}_1 - \mathbf{S}_2 \right\|_F, \tag{14}$$

*and*

$$\left\| \mathbf{D}_1^{-1}\mathbf{S}_1 - \mathbf{D}_2^{-1}\mathbf{S}_2 \right\|_2 \leq \left( \frac{1}{c} + \frac{1}{c^2} N^2 \right) \left\| \mathbf{S}_1 - \mathbf{S}_2 \right\|_F. \tag{15}$$

*Proof.* It can be observed that

$$\begin{aligned}
\left\| \mathbf{D}_1^{-1} - \mathbf{D}_2^{-1} \right\|_F &= \sqrt{\sum_{i=1}^{N} \left( \frac{[\mathbf{D}_1]_{ii} - [\mathbf{D}_2]_{ii}}{[\mathbf{D}_1]_{ii}\,[\mathbf{D}_2]_{ii}} \right)^2} \\
&\leq \frac{1}{c^2} \sqrt{\sum_{i=1}^{N} \left( N \max_{j=1,\ldots,N} \left| [\mathbf{S}_1]_{ij} - [\mathbf{S}_2]_{ij} \right| \right)^2} \\
&\leq \frac{1}{c^2} N \left\| \mathbf{S}_1 - \mathbf{S}_2 \right\|_F,
\end{aligned}$$

which proves Eq. (14). Based on Eq. (14), we further have

$$\begin{aligned}
\left\| \mathbf{D}_1^{-1}\mathbf{S}_1 - \mathbf{D}_2^{-1}\mathbf{S}_2 \right\|_2 &\leq \left\| \mathbf{D}_1^{-1}\mathbf{S}_1 - \mathbf{D}_2^{-1}\mathbf{S}_2 \right\|_F \\
&\leq \left\| \mathbf{D}_1^{-1}\mathbf{S}_1 - \mathbf{D}_1^{-1}\mathbf{S}_2 + \mathbf{D}_1^{-1}\mathbf{S}_2 - \mathbf{D}_2^{-1}\mathbf{S}_2 \right\|_F \\
&\leq \left\| \mathbf{D}_1^{-1} \right\|_2 \left\| \mathbf{S}_1 - \mathbf{S}_2 \right\|_F + \left\| \mathbf{S}_2 \right\|_2 \left\| \mathbf{D}_1^{-1} - \mathbf{D}_2^{-1} \right\|_F \\
&\leq \left\| \mathbf{D}_1^{-1} \right\|_2 \left\| \mathbf{S}_1 - \mathbf{S}_2 \right\|_F + N \left\| \mathbf{D}_1^{-1} - \mathbf{D}_2^{-1} \right\|_F \\
&\leq \left( \frac{1}{c} + \frac{1}{c^2} N^2 \right) \left\| \mathbf{S}_1 - \mathbf{S}_2 \right\|_F,
\end{aligned}$$

through which the proof is completed. $\qquad\square$

Based on Lemma 9 and Lemma 11, the second term in Eq. (13), i.e., $\lambda \left\| \mathbf{H}\mathbf{H}^\top \right\|_2 \left\| \mathbf{D}_1^{-1} - \mathbf{D}_2^{-1} \right\|_F$, can be upperbounded as follows:

$$\lambda \left\| \mathbf{H}\mathbf{H}^\top \right\|_2 \left\| \mathbf{D}_1^{-1} - \mathbf{D}_2^{-1} \right\|_F \leq \frac{\lambda}{c^2} N^2 B^2 \left\| \mathbf{S}_1 - \mathbf{S}_2 \right\|_F. \tag{16}$$

For the third term in Eq. (13), we have

$$\begin{aligned}
&\lambda\sqrt{N} \left\| \mathbf{D}_1^{-1}\mathbf{S}_1\mathbf{H}\mathbf{H}^\top\mathbf{D}_1^{-1} - \mathbf{D}_2^{-1}\mathbf{S}_2\mathbf{H}\mathbf{H}^\top\mathbf{D}_2^{-1} \right\|_F \\
=&\lambda\sqrt{N} \left\| \mathbf{D}_1^{-1}\mathbf{S}_1\mathbf{H}\mathbf{H}^\top\mathbf{D}_1^{-1} - \mathbf{D}_1^{-1}\mathbf{S}_1\mathbf{H}\mathbf{H}^\top\mathbf{D}_2^{-1} + \mathbf{D}_1^{-1}\mathbf{S}_1\mathbf{H}\mathbf{H}^\top\mathbf{D}_2^{-1} - \mathbf{D}_2^{-1}\mathbf{S}_2\mathbf{H}\mathbf{H}^\top\mathbf{D}_2^{-1} \right\|_F \\
\leq&\lambda\sqrt{N} \left\| \mathbf{D}_1^{-1}\mathbf{S}_1\mathbf{H}\mathbf{H}^\top \left( \mathbf{D}_1^{-1} - \mathbf{D}_2^{-1} \right) \right\|_F + \lambda\sqrt{N} \left\| \left( \mathbf{D}_1^{-1}\mathbf{S}_1 - \mathbf{D}_2^{-1}\mathbf{S}_2 \right)\mathbf{H}\mathbf{H}^\top\mathbf{D}_2^{-1} \right\|_F \\
\leq&\lambda\sqrt{N} \left\| \mathbf{D}_1^{-1}\mathbf{S}_1 \right\|_2 \left\| \mathbf{H}\mathbf{H}^\top \right\|_F \left\| \mathbf{D}_1^{-1} - \mathbf{D}_2^{-1} \right\|_2 + \lambda\sqrt{N} \left\| \mathbf{D}_1^{-1}\mathbf{S}_1 - \mathbf{D}_2^{-1}\mathbf{S}_2 \right\|_2 \left\| \mathbf{H}\mathbf{H}^\top \right\|_F \left\| \mathbf{D}_2^{-1} \right\|_2.
\end{aligned}$$

Based on Lemma 9, Lemma 10, and Lemma 11, we can get the following result:

$$\lambda\sqrt{N}\left\|\mathbf{D}_1^{-1}\mathbf{S}_1\mathbf{H}\mathbf{H}^\top\mathbf{D}_1^{-1} - \mathbf{D}_2^{-1}\mathbf{S}_2\mathbf{H}\mathbf{H}^\top\mathbf{D}_2^{-1}\right\|_{\mathrm{F}} \le \left(1+\frac{2}{c}N^2\right)\frac{\lambda}{c^2}N\sqrt{N}B^2\left\|\mathbf{S}_1-\mathbf{S}_2\right\|_{\mathrm{F}}. \quad (17)$$

Substituting the results in Eq. (16) and Eq. (17) into Eq. (13) gives

$$\|\nabla f_S(\mathbf{S}_1) - \nabla f_S(\mathbf{S}_2)\|_{\mathrm{F}} \le \left(2\gamma + 2\mu_2 + \frac{\lambda}{c^2}N^2B^2 + \left(1+\frac{2}{c}N^2\right)\frac{\lambda}{c^2}N\sqrt{N}B^2\right)\|\mathbf{S}_1-\mathbf{S}_2\|_{\mathrm{F}}$$

$$\le \left(2\gamma + 2\mu_2 + \left(1+\frac{1}{c}N\sqrt{N}\right)\frac{2\lambda}{c^2}N^2B^2\right)\|\mathbf{S}_1-\mathbf{S}_2\|_{\mathrm{F}}.$$

Therefore, function $f_S(\mathbf{S})$ is $L$-smooth with $L_S = 2\gamma + 2\mu_2 + \frac{2\lambda}{c^2}N^2B^2 + \frac{2\lambda}{c^3}N^3\sqrt{N}B^2$ and the proof is completed. □

Based on the results in Lemma 6 and Lemma 8, we can conclude that $f_H(\mathbf{H})$ and $f_S(\mathbf{S})$ are both $L$-smooth. To ensure the monotonically decreasing property of (ASMP), the step sizes must satisfy (Parikh and Boyd, 2014):

$$0 < \eta_1 < \frac{2}{L_H} = \frac{1}{1+2\lambda} \quad \text{and} \quad 0 < \eta_2 < \frac{2}{L_S} = \frac{1}{\gamma + \mu_2 + \left(1 + \frac{1}{c}N\sqrt{N}\right)\frac{\lambda}{c^2}N^2B^2}.$$

Under such condition, the convergence of (ASMP) to a first-order stationary point of Problem (4) can be readily obtained based on the results for alternating proximal gradient descent method in Bolte et al. (2014); Nikolova and Tan (2017) with convergence rate

$$\inf_{k\ge K}\left\{\left\|\mathbf{H}^{(k+1)} - \mathbf{H}^{(k)}\right\|_{\mathrm{F}}^2 + \left\|\mathbf{S}^{(k+1)} - \mathbf{S}^{(k)}\right\|_{\mathrm{F}}^2\right\} \le \frac{1}{\rho K}\left(p\big(\mathbf{H}^{(0)},\mathbf{S}^{(0)}\big) - p\big(\mathbf{H}^*,\mathbf{S}^*\big)\right), \quad (18)$$

where $\rho = \min\left\{\frac{1}{\eta_1} - \frac{L_H}{2}, \frac{1}{\eta_2} - \frac{L_S}{2}\right\}$ and $p(\mathbf{H},\mathbf{S})$ represents the objective function in Eq. (4).

## D DISCUSSION ON THE JOINT OPTIMIZATION APPROACH

In this paper, we have used the alternating optimization approach to induce the ASMP scheme, while another natural idea is to apply a joint optimization approach for Problem (4). In this section, we will show that the joint optimization approach actually is inferior compared to the alternating one, since joint optimization would lead to slower convergence, which motivates the use of alternating optimization in ASMP.

We define the smooth part of the objective in Problem (4) as

$$f(\mathbf{H},\mathbf{S}) = \|\mathbf{H} - \mathbf{X}\|_{\mathrm{F}}^2 + \gamma\|\mathbf{S} - \mathbf{A}\|_{\mathrm{F}}^2 + \lambda\mathrm{Tr}\left(\mathbf{H}^\top\mathbf{L}_{\mathrm{rw}}\mathbf{H}\right) + \mu_2\|\mathbf{S}\|_{\mathrm{F}}^2.$$

The $L$-smoothness of $f(\mathbf{H},\mathbf{S})$ is deomnstrated in the following lemma.

**Lemma 12.** *Function $f(\mathbf{H},\mathbf{S})$ is $L$-smooth with $L = \max\Big\{\sqrt{L_H^2 + (1+\frac{1}{c}N\sqrt{N})^2\frac{4\lambda^2}{c^2}NB^2},$*

*$\sqrt{L_S^2 + (1+\frac{1}{c}N^2)^2\frac{4\lambda^2}{c^2}NB^2}\Big\}$, i.e., for any $\mathbf{H}_1,\mathbf{H}_2 \in \mathbb{R}^{N\times M}$ and $\mathbf{S}_1,\mathbf{S}_2 \in \mathbb{R}^{N\times N}$, the following inequality holds:*

$$\|\nabla f(\mathbf{H}_1,\mathbf{S}_1) - \nabla f(\mathbf{H}_2,\mathbf{S}_2)\|_{\mathrm{F}} \le L\sqrt{\|\mathbf{H}_1-\mathbf{H}_2\|_{\mathrm{F}}^2 + \|\mathbf{S}_1-\mathbf{S}_2\|_{\mathrm{F}}^2}.$$

*Proof.* Observe that

$$\left\|\begin{bmatrix}\nabla_\mathbf{H}f(\mathbf{H}_1,\mathbf{S}_1)\\\nabla_\mathbf{S}f(\mathbf{H}_1,\mathbf{S}_1)\end{bmatrix} - \begin{bmatrix}\nabla_\mathbf{H}f(\mathbf{H}_2,\mathbf{S}_2)\\\nabla_\mathbf{S}f(\mathbf{H}_2,\mathbf{S}_2)\end{bmatrix}\right\|_{\mathrm{F}}^2$$

$$= \|\nabla_\mathbf{H}f(\mathbf{H}_1,\mathbf{S}_1) - \nabla_\mathbf{H}f(\mathbf{H}_2,\mathbf{S}_2)\|_{\mathrm{F}}^2 + \|\nabla_\mathbf{S}f(\mathbf{H}_1,\mathbf{S}_1) - \nabla_\mathbf{S}f(\mathbf{H}_2,\mathbf{S}_2)\|_{\mathrm{F}}^2$$

$$= \|\nabla_\mathbf{H}f(\mathbf{H}_1,\mathbf{S}_1) - \nabla_\mathbf{H}f(\mathbf{H}_1,\mathbf{S}_2) + \nabla_\mathbf{H}f(\mathbf{H}_1,\mathbf{S}_2) - \nabla_\mathbf{H}f(\mathbf{H}_2,\mathbf{S}_2)\|_{\mathrm{F}}^2 \quad (19)$$

$$\quad + \|\nabla_\mathbf{S}f(\mathbf{H}_1,\mathbf{S}_1) - \nabla_\mathbf{S}f(\mathbf{H}_2,\mathbf{S}_1) + \nabla_\mathbf{S}f(\mathbf{H}_2,\mathbf{S}_1) - \nabla_\mathbf{S}f(\mathbf{H}_2,\mathbf{S}_2)\|_{\mathrm{F}}^2$$

$$\le L_H^2\|\mathbf{H}_1-\mathbf{H}_2\|_{\mathrm{F}}^2 + L_S^2\|\mathbf{S}_1-\mathbf{S}_2\|_{\mathrm{F}}^2$$

$$\quad + \|\nabla_\mathbf{H}f(\mathbf{H}_1,\mathbf{S}_1) - \nabla_\mathbf{H}f(\mathbf{H}_1,\mathbf{S}_2)\|_{\mathrm{F}}^2 + \|\nabla_\mathbf{S}f(\mathbf{H}_1,\mathbf{S}_1) - \nabla_\mathbf{S}f(\mathbf{H}_2,\mathbf{S}_1)\|_{\mathrm{F}}^2,$$

where the Lipschitz constants $L_H$ and $L_S$ are given in Lemma 6 and Lemma 8.

In the following, we will derive the upper bound for the third and the fourth term in Eq. (19). Based on Lemma 11, the term $\|\nabla_{\mathbf{H}} f(\mathbf{H}_1, \mathbf{S}_1) - \nabla_{\mathbf{H}} f(\mathbf{H}_1, \mathbf{S}_2)\|_{\mathrm{F}}$ can be upperbounded as follows:

$$\|\nabla_{\mathbf{H}} f(\mathbf{H}_1, \mathbf{S}_1) - \nabla_{\mathbf{H}} f(\mathbf{H}_1, \mathbf{S}_2)\|_{\mathrm{F}}$$

$$= \left\| 2\lambda \left( \mathbf{I} - \mathbf{D}_1^{-1} \mathbf{S}_1 \right) \mathbf{H}_1 - 2\lambda \left( \mathbf{I} - \mathbf{D}_2^{-1} \mathbf{S}_2 \right) \mathbf{H}_1 \right\|_{\mathrm{F}} \tag{20}$$

$$\leq 2\lambda \left\| \mathbf{D}_1^{-1} \mathbf{S}_1 - \mathbf{D}_2^{-1} \mathbf{S}_2 \right\|_2 \|\mathbf{H}_1\|_{\mathrm{F}}$$

$$\leq \left(1 + \frac{1}{c} N^2\right) \frac{2\lambda}{c} \sqrt{N} B \|\mathbf{S}_1 - \mathbf{S}_2\|_{\mathrm{F}}. \tag{21}$$

Besides, for the forth term, we have

$$\|\nabla_{\mathbf{S}} f(\mathbf{H}_1, \mathbf{S}_1) - \nabla_{\mathbf{S}} f(\mathbf{H}_2, \mathbf{S}_1)\|_{\mathrm{F}}$$

$$\leq \left\| \lambda \mathbf{D}_1^{-1} \left( \mathbf{H}_1 \mathbf{H}_1^\top - \mathbf{H}_2 \mathbf{H}_2^\top \right) - \lambda \mathrm{Diag} \left( \mathbf{D}_1^{-1} \mathbf{S}_1 \mathbf{H}_1 \mathbf{H}_1^\top \mathbf{D}_1^{-1} - \mathbf{D}_1^{-1} \mathbf{S}_1 \mathbf{H}_2 \mathbf{H}_2^\top \mathbf{D}_1^{-1} \right) \mathbf{1}^\top \right\|_{\mathrm{F}}$$

$$\leq \lambda \left\| \mathbf{D}_1^{-1} \right\|_2 \left\| \mathbf{H}_1 \mathbf{H}_1^\top - \mathbf{H}_2 \mathbf{H}_2^\top \right\|_{\mathrm{F}} + \lambda \sqrt{N} \left\| \mathbf{D}_1^{-1} \mathbf{S}_1 \right\|_2 \left\| \mathbf{H}_1 \mathbf{H}_1^\top - \mathbf{H}_2 \mathbf{H}_2^\top \right\|_{\mathrm{F}} \left\| \mathbf{D}_1^{-1} \right\|_2.$$

According to Lemma 10, we have

$$\|\nabla_{\mathbf{S}} f(\mathbf{H}_1, \mathbf{S}_1) - \nabla_{\mathbf{S}} f(\mathbf{H}_2, \mathbf{S}_1)\|_{\mathrm{F}} \leq \left(1 + \frac{1}{c} N \sqrt{N}\right) \frac{\lambda}{c} \left\| \mathbf{H}_1 \mathbf{H}_1^\top - \mathbf{H}_2 \mathbf{H}_2^\top \right\|_{\mathrm{F}}.$$

Then, we can get

$$\|\nabla_{\mathbf{S}} f(\mathbf{H}_1, \mathbf{S}_1) - \nabla_{\mathbf{S}} f(\mathbf{H}_2, \mathbf{S}_1)\|_{\mathrm{F}}$$

$$= \left(1 + \frac{1}{c} N \sqrt{N}\right) \frac{\lambda}{c} \left\| \mathbf{H}_1 \mathbf{H}_1^\top - \mathbf{H}_1 \mathbf{H}_2^\top + \mathbf{H}_1 \mathbf{H}_2^\top - \mathbf{H}_2 \mathbf{H}_2^\top \right\|_{\mathrm{F}}$$

$$\leq \left(1 + \frac{1}{c} N \sqrt{N}\right) \frac{\lambda}{c} \left( \|\mathbf{H}_1\|_2 \|\mathbf{H}_1 - \mathbf{H}_2\|_{\mathrm{F}} + \|\mathbf{H}_1 - \mathbf{H}_2\|_{\mathrm{F}} \|\mathbf{H}_2\|_2 \right) \tag{22}$$

$$\leq \left(1 + \frac{1}{c} N \sqrt{N}\right) \frac{\lambda}{c} \left( \|\mathbf{H}_1\|_{\mathrm{F}} + \|\mathbf{H}_2\|_{\mathrm{F}} \right) \|\mathbf{H}_1 - \mathbf{H}_2\|_{\mathrm{F}}$$

$$\leq \left(1 + \frac{1}{c} N \sqrt{N}\right) \frac{2\lambda}{c} \sqrt{N} B \|\mathbf{H}_1 - \mathbf{H}_2\|_{\mathrm{F}}.$$

Substituting the results in Eq. (21) and Eq. (22) into Eq. (19) gives

$$\left\| \begin{bmatrix} \nabla_{\mathbf{H}} f(\mathbf{H}_1, \mathbf{S}_1) \\ \nabla_{\mathbf{S}} f(\mathbf{H}_1, \mathbf{S}_1) \end{bmatrix} - \begin{bmatrix} \nabla_{\mathbf{H}} f(\mathbf{H}_2, \mathbf{S}_2) \\ \nabla_{\mathbf{S}} f(\mathbf{H}_2, \mathbf{S}_2) \end{bmatrix} \right\|_{\mathrm{F}}^2$$

$$\leq \left( L_H^2 + \left(1 + \frac{1}{c} N \sqrt{N}\right)^2 \frac{4\lambda^2}{c^2} N B^2 \right) \|\mathbf{H}_1 - \mathbf{H}_2\|_{\mathrm{F}}^2 + \left( L_S^2 + \left(1 + \frac{1}{c} N^2\right)^2 \frac{4\lambda^2}{c^2} N B^2 \right) \|\mathbf{S}_1 - \mathbf{S}_2\|_{\mathrm{F}}^2.$$

Thus, function $f(\mathbf{H}, \mathbf{S})$ is $L$-smooth with

$$L = \max \left\{ \sqrt{L_H^2 + \left(1 + \frac{1}{c} N \sqrt{N}\right)^2 \frac{4\lambda^2}{c^2} N B^2}, \sqrt{L_S^2 + \left(1 + \frac{1}{c} N^2\right)^2 \frac{4\lambda^2}{c^2} N B^2} \right\}, \tag{23}$$

through which the proof is completed. □

This result in Lemma 12 indicates that the Lipschitz constant $L$ is larger than both $L_H$ and $L_S$. Practical graphs are commonly large graphs; i.e., the number of nodes $N$ would dominate the other constants in $L$, i.e., $c$, $\lambda$, and $B$. Therefore, if we use the joint optimization approach, the Lipschitz constant is larger than $L_H$ and $L_S$ by a large margin.

After deriving the Lipschitz constant in joint optimization, we compare its convergence rate with the alternating optimization approach. Denote $\{\mathbf{H}^{(k)}, \mathbf{S}^{(k)}\}_{k=0}^K$ as the sequence generated by the above joint optimization approach. Following the results in Bolte et al. (2014); Nikolova and Tan (2017), the convergence property of the joint optimization approach is also given in Eq. (18) with $\rho = \frac{1}{\eta} - \frac{L}{2}$. Theoretically, to guarantee the sufficient descent of the objective at each step, $\rho$ can be chosen to be $\min\{\frac{1}{\eta_1} - \frac{L_H}{2}, \frac{1}{\eta_2} - \frac{L_S}{2}\}$ in the alternating optimization approach. Due to the fact that $L > \max\{L_H, L_S\}$ according to Eq. (23), the alternating optimization approach is allowed to adopt a larger step size at each block than the joint optimization approach, resulting in a faster convergence behavior of the sequence. Motivated by this fact, we develop ASMP based on the alternating procedure rather than the joint one so that the resulting message passing structure contains fewer layers to achieve the similar or even better numerical performance compared to the joint one.

# E  ADDITIONAL EXPERIMENTS

## E.1  RUNTIME ANALYSIS

The computation complexity of ASMP is in order $\mathcal{O}\left(N^2 d\right)$, where $N$ is the number of nodes and $d$ is the feature dimension. This is larger than the complexity of simple graph convolution, which is in order $\mathcal{O}\left(Ed\right)$ with $E$ being the number of edges. To better understand the complexity of ASGNN and other baselines, we provide a runtime comparison of different models. Specifically, we train different models for 200 epochs on Cora dataset under nettack with one perturbation per target node and the averaged runtimes from ten trials are shown in the following table.

Table 4: Runtime (in seconds) for 200 epochs of training.

| Model | GCN | GCN-Jaccard | GCN-SVD | Pro-GNN | GNNGuard | Elastic GNN | ASGNN |
|---|---|---|---|---|---|---|---|
| Time | 1.44 | 2.00 | 1.56 | 193.34 | 36.68 | 28.89 | 42.06 |

From the table, we can see that although ASGNN requires more computation time than GCN, GCN-Jaccard, and GCN-SVD, it is much faster than Pro-GNN and the complexity of ASGNN is comparable to GNNGuard and Elastic GNN.

## E.2  THE LEARNED COEFFICIENTS IN ASGNN

To better understand how different terms in the joitn node feature learning and graph structure learning objective, we provide the learned ASGNN coefficients on Cora dataset under targeted attack below.

Table 5: Learned coefficients of ASGNN on Cora under targeted attack

| Ptb. number | $\lambda$ | $\gamma$ | $\mu_1$ | $\mu_2$ | $\eta_1$ | $\eta_2$ |
|---|---|---|---|---|---|---|
| 0 | 8.1085 | 0.1674 | 0.3808 | 0.4359 | 0.0441 | 0.0035 |
| 1 | 8.3985 | 0.3440 | 0.1673 | 0.3802 | 0.0365 | 0.0026 |
| 2 | 8.3891 | 0.3263 | 0.2060 | 0.3990 | 0.0382 | 0.0025 |
| 3 | 8.4248 | 0.1972 | 0.2003 | 0.3442 | 0.0369 | 0.0023 |
| 4 | 8.3735 | 0.0908 | 0.2199 | 0.4070 | 0.0368 | 0.0017 |
| 5 | 8.4753 | 0.1246 | 0.2005 | 0.3061 | 0.0381 | 0.0029 |

From the table, we can see that the value of $\lambda$ is much larger than other coefficients. Besides, the value of the coefficients do not vary much under different perturbation numbers, indicating the importance of different terms are not affected by the perturbation number.

## E.3  SPARSITY LEVEL OF THE LEARNED GRAPHS IN ASGNN

The sparsity level of the graphs (percentage of zero elements in the learned adjacency matrix) generated in the last layer of ASGNN are provided below. Note that the sparsity level can be tuned by constraining the learning space of $\mu_1$. For example, we can fix $\mu_1$ to be a large value, which will lead to sparser graphs.

Table 6: Sparsity level of the graphs leant in ASGNN.

| Dataset | Cora | Citeseer | Cora-ML | ACM |
|---|---|---|---|---|
| Sparsity level (%) | $77.27 \pm 18.39$ | $89.40 \pm 0.95$ | $78.54 \pm 12.53$ | $46.56 \pm 3.24$ |

## E.4  CONVERGENCE PROPERTY OF ASMP IN PRACTICE

According to Theorem 4, the convergence of ASMP is guaranteed with proper choices of step sizes. Since the step sizes are set to be learnable parameters, we provide an additional experiment to evaluate the convergence of ASMP with learned step sizes empirically.

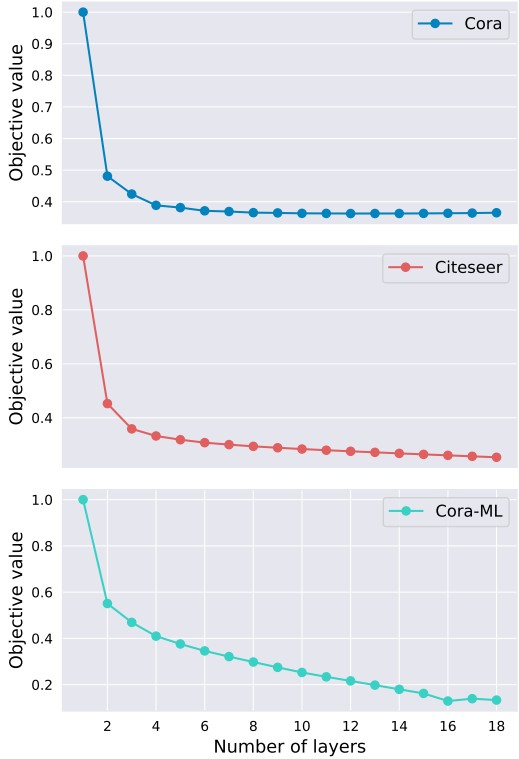

Figure 2: The objective value in Problem (4) during ASMP.

To evaluate the convergence property of ASMP with learned step sizes, we conduct experiments on Cora, Citeseer, and Cora-ML datasets at a 25% perturbation rate under meta-attack. Specially, we train a 4-layer ASGNN model and observe that the learned step sizes do not satisfy the condition in Theorem 4. In the following, we investigate the empirical convergence property of ASMP. Since we use a recurrent structure in ASGNN, i.e., the step sizes used in different layers are the same, we are able to extend the trained 4-layer ASGNN model to a deeper one. The values of the objective function in Problem (4) in different layers are showcased in Figure 2, where the objective values are normalized by dividing the objective value in the first layer. From Figure 2, we can find that ASMP with learned step sizes can monotonically decrease the objective function value during the message passing process. Note that although the monotonic decreasing property does not hold in 16-18 layers on the Cora-ML dataset, this may be mainly due to the fact that the step sizes are learned only based on a 4-layer model. The results indicate that although the learned step sizes do not satisfy the condition in Theorem 4, they still ensure the monotonic decrease of the objective function value in practice.

