# OpenReview forum: "ASGNN: Graph Neural Networks with Adaptive Structure"
_ICLR.cc/2023/Conference — Submitted to ICLR 2023_

### Official Review · Reviewer_Qd9P · 2022-10-24

**Confidence:** 4
**Correctness:** 4
**Technical Novelty And Significance:** 3
**Empirical Novelty And Significance:** 3
**Recommendation:** 6

**Clarity, Quality, Novelty And Reproducibility:**

**Clarity**: very good and easy to follow.

**Quality**: There are some important weaknesses of the paper and the experiments:
* The authors do not compare to GNNGuard [Zhang and Zitnik, 2020], which is an important baseline.
* The authors do not evaluate their defense on adaptive attacks, i.e., attacks designed to overcome the specific defense the authors propose. This is very important, as it is always easier to defend against existing attacks, but often an attacker can break novel defenses by adapting the strategy. This is, of course, different from certifiable defenses where we get a mathematical guarantee of robustness.
* There is no information about the computational complexity of the method and/ or runtime analysis. Given that the authors only report on small datasets, I expect the method to be quite computationally expensive.
* From the description in the text it is unclear whether the authors repeated all experiments ten times on one single split of the data or whether they evaluate on ten different splits. If they only evaluate on a single split, this is a problem because GNN performance has been shown to be highly variable depending on the individual split [Shchur et al., 2018].
* It would be interesting to see some qualitative insight into what kind of graphs the method learns. Does it remove adversarial edges? How sparse are the resulting graphs?
* There are no hyperparameter sensitivity of ablation study results. There are five terms in the loss function: are all of them required, and how sensitive is the model to different choices of hyperparameters?

**Novelty**: While the loss function is very similar to the one in Pro GNN [Jin et al. 2020] and the idea of framing message passing as gradient steps is not new, their combination is novel, interesting and non-trivial.

**Reproducibility**: Okay; while the description of the method is sufficiently clear to implement it, in order to exactly reproduce the results some information is missing: the authors did not provide an implementation of their method, did not provide details which of the hyperparameters in their search space was selected in the end, it is unclear which exact split of the data was used, and the hyperparameters of the MLP of the method are not provided.


References:
Shchur, O., Mumme, M., Bojchevski, A., & Günnemann, S. (2018). Pitfalls of graph neural network evaluation. Relational Representation Learning Workshop, NeurIPS 2018.

**Strength And Weaknesses:**

Strengths:
* Framing the structure learning as an optimization approximately solved via the message-passing procedure itself appears novel and is interesting
* The presented results suggest that the method is effective at defending against existing attack methods.
* The presentation of the work is clear and easy to follow.


Weaknesses:
* The authors do not evaluate their defense against adaptive attacks designed to circumvent their defense
* The comparison is missing an important baseline with GNNGuard [Zhang and Zitnik, 2020].
* The authors only evaluate their method on small datasets.

**Summary Of The Paper:**

The authors propose ASGNN, a robust GNN architecture. ASGNN learns a "cleaned" adjacency matrix as well as feature matrix per message passing layer with the goal of removing adversarial perturbations from the graph. The message passing procedure is framed as an optimization problem, and the individual message passing steps are proximal gradient steps. The authors present experimental results on well-known datasets from the literature.

**Summary Of The Review:**

The paper is a solid piece of work developing a robust GNN architecture. The presentation is clear and easy to follow. There are some key limitations of the work that prevent me from recommending acceptance (see details above):
* Missing the GNNGuard baseline
* Lack of adaptive attacks.
* Only reports results on small datasets, no runtime/ complexity information or ablation study results provided.

---

> ### Author Response · Authors · 2022-11-19
> **Answer to Reviewer Qd9P (Part 1)**
>
> > Q1: The authors do not evaluate their defense against adaptive attacks
> designed to circumvent their defense.
>
> We appreciate the reviewer for his/her valuable suggestions. In this paper, the experimental setting of the adversarial attacks exactly follows that of the existing work, (e.g., the one in
> Pro-GNN). We noticed there is an interesting recent work proposing to use adaptive attack
> [1], which is released after this ICLR submission deadline. The evaluation under adaptive attacks is indeed an important problem, however, due to the time limit, we cannot investigate this during this rebuttal period, but we are happy to include the relevant experiments in the future.
>
> > Q2: The comparison is missing an important baseline with GNNGuard.
>
> Thank you for pointing this out. We have added this baseline to the experiments in the revised paper.
>
> > Q3: There is no information about the computational complexity of the
> method and/or runtime analysis. Given that the authors only report
> on small datasets, I expect the method to be quite computationally
> expensive.
>
> Thanks very much for this suggestion. The computation complexity of ASMP is
>  $\mathcal{O}\left(N^{2}d\right)$, where $N$ is the number
> of nodes and $d$ is the feature dimension. We also provide a runtime comparison of different models. We train different models for 200 epochs on the Cora
> dataset under nettack with one perturbation per target node. The averaged
> runtimes from ten trials are shown as follows:
>
> | Model | GCN  | GCN-Jaccard | GCN-SVD | Pro-GNN | GNNGuard | Elastic GNN | ASGNN |
> |:-----------:|:------:|:-------------:|:--------:|:---------:|:----------:|:-------------:|:-------:|
> | Time (seconds)  | 1.44 | 2.00        | 1.56    | 193.34  | 36.68    | 28.89       | 42.06 |
>
> From the table, comparing the computation time, we can see that ASGNN is more complex than GCN, GCN-Jaccard, and GCN-SVD and is comparable to GNNGuard and Elastic GNN. But ASGNN is much faster than Pro-GNN.
>
> > Q4: From the description in the text it is unclear whether the authors
> repeated all experiments ten times on one single split of the data
> or whether they evaluate on ten different splits.
>
> To keep the results comparable, we follow the data split strategy used in Pro-GNN, which repeats the experiments on one single split. Conducting experiments on different data splits is indeed an important problem to investigate, however, due to the time limit, we fail to finish this during the rebuttal. But we will add this experiment to the final version.
>
> > Q5: It would be interesting to see some qualitative insight into what
> kind of graphs the method learns. Does it remove adversarial edges?
> How sparse are the resulting graphs?
>
> Thank you for this valuable suggestion. Since the graph without attacks
> may also contain task-irrelevant edges, it is generally hard to say
> what kind of graphs are the best for the downstream tasks. Thus, there is no unique and accurate metric to measure the quality of the learned graph.
> In the following, we will train a GCN model using the graph generated in the last layer of ASGNN, which we named GCN-AS.
> Under the assumption that the GCN trained on a better graph will give better performance, the quality of the learned graph in ASGNN can be measured by evaluating the performance of GCN-AS.
> Specifically, we conduct experiments on different datasets under targeted attack with five perturbations per node and the results are shown below.
>
>
> | Model  | Cora | Citeseer | Cora-ML | ACM |
> |:--------:|:------:|:----------:|:--------:|:-----:|
> | GCN    |  56.75$\pm$1.37   |  52.70$\pm$1.98        |    64.52$\pm$1.29     | 73.32$\pm$1.77    |
> | GCN-AS |   67.91$\pm$9.28   |   69.96$\pm$5.30       |  79.63$\pm$7.98       | 87.56$\pm$1.49    |
>
> It can be seen that the performance of GCN-AS is much better than
> GCN, which indicates the learned graph of ASGNN can mitigate the influence
> of the adversarial attacks. Moreover, GCN-AS even outperforms other
> defense models on some datasets, e.g., GCN-AS outperforms all other
> baselines on the ACM dataset.
>
> The sparsity level of the graphs (percentage of zero elements in the
> learned adjacency matrix) generated in the last layer of ASGNN are
> as follows:
> | Dataset  | Cora | Citeseer | Cora-ML | ACM |
> |:--------:|:------:|:----------:|:--------:|:-----:|
> | Sparsity level (%)    |  77.27$\pm$18.39   |  89.40$\pm$0.95        |    78.54$\pm$12.53     | 46.56$\pm$3.24    |
>
> Note that the sparsity level can be tuned by constraining the learning
> space of $\mu_{1}$. For example, we can fix $\mu_{1}$ to be a large
> value, which will lead to sparser graphs.

---

> > ### Author Response · Authors · 2022-11-19
> > **Answer to Reviewer Qd9P (Part 2)**
> >
> > > Q6: There are no hyperparameter sensitivity of ablation study results. There are five terms in the loss function: are all of them required, and how sensitive is the model to different choices of hyperparameters?
> >
> > Thanks for this question. In this paper, we design the ASGNN model based on an iterative optimization algorithm. The five different terms in the optimization objective carry different optimization targets with associated coefficients controlling the relative importance of these terms.
> > Actually, the coefficients are treated as learnable parameters instead of
> > hyperparameters.
> > For example, the learned coefficients on the Cora dataset
> > under targeted attack are given in Appendix E.2 in the revised paper.
> >
> > > Q7: the authors did not provide an implementation of their method.
> >
> > In this revision, we have added an implementation of our method in the supplementary material.
> >
> > **Reference**
> >
> >  [1] Are Defenses for Graph Neural Networks Robust? Felix Mujkanovic, Simon Geisler, Stephan Günnemann, and Aleksandar Bojchevski. In Neural Information Processing Systems (NeurIPS), 2022

---

> > > ### Comment · Reviewer_Qd9P · 2022-11-25
> > > **Response**
> > >
> > > Dear authors,
> > >
> > > Thank you for your efforts addressing my comments. I have some follow-up comments/ questions:
> > >
> > > > The computation complexity of ASMP is $O(N^2d)$
> > >
> > > As the method learns a different adjacency matrix per message-passing layer, should this not also depend on the number of layers?
> > >
> > > Regarding the second table in your response, this is very interesting (GCN vs GCN-AS). The results appear convincing, however the "error bars" are quite large. Do you have an intuition for why this is the case?
> > >
> > > The same is true for the sparsity results. Moreover, it would be interesting to compare the sparsity with the sparsity of the original graphs to see whether the method increases or decreases the sparsity.
> > >
> > > All in all, my comments were adequately addressed and I have increased my score.

---

> > > > ### Author Response · Authors · 2022-12-06
> > > > **Response to Reviewer Qd9P**
> > > >
> > > > We would like to thank the reviewer for his/her detailed comments and valuable feedback. We answer the questions below:
> > > > > Q1: The computation complexity of ASMP is $\mathcal{O}(N^{2}d)$. As the method learns a different adjacency matrix per message-passing layer, should this not also depend on the number of layers?
> > > >
> > > > Thank you for pointing this out. The computation complexity of a layer in ASMP mainly comes from the computation of $\mathbf{S}\mathbf{H}\mathbf{H}^{T}$, which is in order $\mathcal{O}(N^{2}d)$. Thus, the computation complexity of ASMP should be $\mathcal{O}(KN^{2}d)$, where K is the number of layers of ASMP.
> > > >
> > > > > Q2: The results appear convincing, however the "error bars" are quite large. Do you have an intuition for why this is the case?
> > > >
> > > > Thanks for this comment. We think the reason why the error bars are quite large possibly comes from the fact that the performance of GCN-AS is not explicitly expressed in the learning target of ASGNN. Thus, the graphs generated by ASGNN cannot guarantee a consistently well performance of GCN-AS. The same holds for the sparsity results. An additional possible reason why the error bars of the sparsity results are quite large in some datasets is that the sparsity level is controlled by the coefficients in the joint node feature and graph structure learning objective, which are learned through minimizing the loss function. Since the model parameters may vary a lot with different initializations, the sparsity level of the learned graph in ASGNN is quite random.
> > > >
> > > > > Q3: Moreover, it would be interesting to compare the sparsity with the sparsity of the original graphs to see whether the method increases or decreases the sparsity.
> > > >
> > > > Thanks for the suggestion. The sparsity level of the attacked graphs in Cora, Citeseer, Cora-ML, and ACM datasets is 99.82%, 99.82%, 99.76%, and 99.63%, respectively, which indicates that the learned graph in ASGNN is much denser than the attacked graph. This is mainly because the learned graphs in ASGNN are weighted graphs while the attacked graphs are unweighted. Note that if we want the graphs in ASGNN to be sparser, we can constrain the learning space of the coefficients in the joint node feature and graph structure learning objective, which is an interesting direction to investigate in the future.

---

### Official Review · Reviewer_sBNe · 2022-10-25

**Confidence:** 4
**Clarity, Quality, Novelty And Reproducibility:** 1. Limited innovation.
2. The code is…
**Correctness:** 3
**Technical Novelty And Significance:** 2
**Empirical Novelty And Significance:** 2
**Recommendation:** 5

**Strength And Weaknesses:**

Strength:
1. It’s a generalized framework, achieving conduct message passing over different graph structures at different layers.
2. In most experiments, it achieves the best results.
3. The proposed update scheme has good theoretical support.

Weaknesses:
1. The authors claim that the message passing scheme of ASGNN is ‘interpretable’. But relevant experimental results are not given.
2. This paper lacks a complexity analysis of the proposed model, considering the proposed ASGNN is trained to learn different propagation structures on each GNN layer.
3. The challenges and innovation of introducing the adaptive structure are not clear.



**Summary Of The Paper:**

The major contribution is the message passing scheme with the adaptive structure that enables learning different propagation structures for different GNN layers. The proposed method ASGNN is one of the graph structure learning methods and the improvement of this work in terms of motivation is marginal.


**Summary Of The Review:**

The motivation of this work is not quite novel. And the adaptive structure assumption increases the complexity of the proposed model. In a word, the proposed ASGNN brings performance improvements, but the complexity of this model also makes it hard to apply to large-scale graph data.

---

> ### Author Response · Authors · 2022-11-19
> **Answer to Reviewer sBNe**
>
> > Q1: The authors claim that the message passing scheme of ASGNN is
> ‘interpretable’. But relevant experimental results are not given.
>
> In this paper, we call ASGNN interpretable in the sense that its message passing scheme
> is derived based on the alternating proximal gradient descent algorithm.
> Thus, the message passing process in a trained ASGNN carries out the
> interpretation of a parameter-optimized iterative algorithm for simultaneous
> node feature and graph structure learning.
>
> > Q2: This paper lacks a complexity analysis of the proposed model,
> considering the proposed ASGNN is trained to learn different propagation
> structures on each GNN layer.
>
> Thanks for your comment. The computation complexity of ASMP mainly
> comes from the computation of $\mathbf{S}\mathbf{H}\mathbf{H}^{T}$,
> which is $\mathcal{O}\left(N^{2}d\right)$ (where $N$ is
> the number of nodes and $d$ is the feature dimension). To provide
> a fair comparison of the complexity of different models, we conduct a runtime
> analysis in Appendix E.1 of the revised paper.
>
> > Q3: The challenges and innovation of introducing the adaptive structure
> are not clear.
>
> The target of this paper is to design a robust GNN model. The major novelty of this work lies in the intersection of graph structure learning and optimization-induced GNN. Please note that this work has innovations from both sides and the design of ASGNN is novel, interesting, and non-trivial, which has also been recognized by Reviewer Qd9P.
>
> This work differs from existing works on graph structure learning. Existing ones mainly focus on designing
> **an extra component appended to existing GNN models to improve their robustness**, including
> the pre-processing based methods (such as GCN-Jaccard and GCN-SVD) and
> the alternating optimization based methods (such as Pro-GNN). Different
> from them, the proposed ASGNN model is **an inherently robust GNN architecture**.
> So ASGNN is different from existing graph structure learning methods.
> Thus, the design principle based on an iterative algorithm for robust GNN models presented in this paper is novel.  Moreover, since ASGNN itself is a GNN architecture,
> its performance can be potentially improved with the aforementioned existing graph structure
> learning methods like GNN-Jaccard, GNN-SVD, and Pro-GNN.
>
> Existing optimization-induced GNNs only involve one optimization variable
> and do not consider the optimization of the graph structure. In this
> work, we consider the simultaneous node feature and graph structure
> learning and a major challenge is how to induce a GNN model from this joint optimization problem. To tackle this problem, a novel alternating optimization algorithm is proposed for
> problem resolution.
> The alternating optimization algorithm is also shown to be more effective than the joint optimization approach. The proposed alternating algorithm is not only
> friendly to back propagation but also provably convergent.
>
> > Q4: The code is not open, and some setting of hyperparemeters like
> $\mu_{1}$, $\mu_{2}$ is unknown.
>
> As to the opening of the code, in this revision, we have added the implementation details of our method in the Supplementary Materials.
>
> As to the ``setting of hyperparameters'', we would like to clarify that coefficients $\mu_{1}$, $\mu_{2}$ in ASGNN are treated as learnable parameters but not hyperparameters.
> For example, the learned values of $\mu_{1}$ and $\mu_{2}$ on the Cora dataset
> under targeted attack are given in Appendix E.2 of the revised paper.

---

> ### Author Response · Authors · 2022-12-13
> **Reply to Reviewer sBNe**
>
> We appreciate the reviewer for his/her comments, which greatly help us improve the clarity and scope of our paper. We believe our response and the additional results in the revision have addressed your concerns. We sincerely hope the reviewer finds our response useful and updates the scores if your concerns have been resolved. Please let us know if you have further questions or concerns.

---

### Official Review · Reviewer_tSZb · 2022-11-02

**Confidence:** 2
**Correctness:** 3
**Technical Novelty And Significance:** 3
**Empirical Novelty And Significance:** 3
**Recommendation:** 6

**Clarity, Quality, Novelty And Reproducibility:**

The paper is well written. I do not know much of the related work. However, it seems the main idea is novel and well-developed.

**Strength And Weaknesses:**

### Strengths:
1. The paper is well-motivated to handle the noise in graph structure with simultaneous learning of graph structure with features.
1. The idea of adaptively learning the graph structure iteratively is appealing.
1. Experimental results show that the method is promising. However, improvements on most datasets is only marginal.

### Weaknesses:
1. I do not understand the usefulness of having ‘different structure’ in each layer. More plausible to me is that the with adaptive graph structure, it gets progressively better and not just different.
1. Why do authors use $L_{rw}$ and not $L_{sym}$. Is there any reason which makes it better? Experimental analysis to study the difference would be helpful as well.
1. The paper can also be strengthened with further ablations. What are the learnt values of ASGNN coefficients which give the best results? This would be helpful to see the terms affecting the performance.


**Summary Of The Paper:**

The paper deals with building robust graph learning methods not susceptible to adversarial attacks. Existing methods learn the structure of graph either by preprocessing the graph structure or by parametrically learning the graph adjacency. In this paper, the authors propose to simultaneously learn the graph structure as well as to use it for message passing iteratively. They call it message passing with adaptive structure (ASMP), which is shown as a way of doing proximal gradient descent for simultaneous denoising of graph signal and graph structure. Experimental results are provided to show the usefulness of the approach.

**Summary Of The Review:**

The overall idea of adaptively learning graph structure with message passing seems nice and well-developed. However, I am not fully aware of the related works and hence, not clear on the degree of novelty and significance of the results. Hence, I'm less confident. Therefore, I'm going with weak acceptance.

---

> ### Author Response · Authors · 2022-11-19
> **Answer to Reviewer tSZb**
>
> > Q1: I do not understand the usefulness of having ‘different structure’ in each layer. More plausible to me is that the with adaptive graph
> structure, it gets progressively better and not just different.
>
> Thank you for this insightful question. Please kindly note that the proposed ASMP scheme is derived from the alternating proximal gradient descent algorithm for an optimization problem of simultaneous node feature and graph structure learning, so the graph structure in ASMP is indeed getting progressively better in the sense that it reduces the objective value. To further illustrate that the learned graph in ASGNN is getting better, we provide an additional experiment in Section 4.3 of the revised paper.
>
> > Q2: Why do authors use $\mathbf{L}\_{\mathrm{rw}}$ and not $\mathbf{L}\_{\mathrm{sym}}$.
> Is there any reason which makes it better?
>
> Yes. Both $\mathbf{L}\_{\mathrm{rw}}$ and $\mathbf{L}\_{\mathrm{sym}}$ are widely used in the GNN literature and also applicable in our model, however, we observe that in our ASGNN model when $\mathbf{L}\_{\mathrm{sym}}$ is used the gradient w.r.t. $\mathbf{S}$ has a more complicated form than that when $\mathbf{L}\_{\mathrm{rw}}$ is used, which will lead to higher computational complexity. To obtain a more efficient model, we finally choose $\mathbf{L}\_{\mathrm{rw}}$.
>
> > Q3: What are the learned values of ASGNN coefficients which give the
> best results? This would be helpful to see the terms affecting the
> performance.
>
> Thanks for the valuable suggestion. To showcase this point, we provide the learned ASGNN coefficients on the Cora dataset under the case of  targeted attack as follows:
> | Ptb. number | $\lambda$   | $\gamma$ | $\mu_1$ | $\mu_2$    |   $\eta_1$     | $\eta_2$       |
> |:-------------:|--------|----------|--------:|--------|--------|--------|
> | 0           | 8.1085 | 0.1674   | 0.3808  | 0.4359 | 0.0441 | 0.0035 |
> | 1           | 8.3985 | 0.3440   | 0.1673  | 0.3802 | 0.0365 | 0.0026 |
> | 2           | 8.3891 | 0.3263   | 0.2060  | 0.3990 | 0.0382 | 0.0025 |
> | 3           | 8.4248 | 0.1972   | 0.2003  | 0.3442 | 0.0369 | 0.0023 |
> | 4           | 8.3735 | 0.0908   | 0.2199  | 0.4070 | 0.0368 | 0.0017 |
> | 5           | 8.4753 | 0.1246   | 0.2005  | 0.3061 | 0.0381 | 0.0029 |
>
>
> From the above table, we can see that the value of $\lambda$ is consistently much larger
> than other coefficients. Besides, the values of these coefficients do not vary too much under different perturbation numbers, indicating that the importance of different terms in the joint node feature and graph structure learning objective is not affected by the perturbation
> numbers.

---

### Author Response · Authors · 2022-11-19
**Common replies to all reviewers**

We would like to thank the reviewers for their efforts in reviewing
our paper and their constructive suggestions on improving our manuscript.
We have added more clarifications and new experiments to address reviewers'
concerns in the updated manuscript. The main changes are listed below:

1. We reorganize some parts of the paper for clarity. For example, we replace Figure 1 and we add some assumptions to the convergence theorem and further state the convergence rate in the theorem. Besides, we rewrite the Appendix for clarity.

2. We add a new baseline model GNNGuard in the experiment.

3. We replace the experiment in Section 4.3 to demonstrate that ASGNN can help purify the perturbed graph.

4. We add a runtime analysis in Appendix E.1.

5. We add a discussion on the learned coefficients in ASGNN in Appendix E.2.

6. We add a discussion on the sparsity level of ASGNN in Appendix E.3.

We hope we have addressed all your concerns. If it is not the case,
we will be happy to further answer any questions and improve the manuscript.

---

### Decision · Program_Chairs · 2023-01-20

**Decision:**

Reject

**Justification For Why Not Higher Score:**

As described above, an accept as poster would be justifiable as well.

**Justification For Why Not Lower Score:**

N/A

**Metareview: Summary, Strengths And Weaknesses:**

The authors tackle robustness of graph neural networks. They propose a novel model which optimizes the adjacency/feature matrix with the aim to remove adversarial changes. From a technical site, the message passing scheme is formulated as an optimization problem which allows the authors to introduce the adaptive graph structure in a principles way. This, indeed, is a nice technical contribution of the paper. The experiments show good performance of the method.

As also highlighted by some other reviewers, the high complexity (quadratic in the number of nodes) might not be practical for real applications -- accordingly, the authors test also only rather small datasets. Moreover, adapting the graph structure to improve robustness has also been explored by multiple other works, thus, limiting a bit the novelty.

Overall, the paper presents a nice solution to an important problem. Weighing pros and cons, it remains a borderline case. Considering the overall set of papers in my batch, I provide 'reject' at the moment, but the decision can be improved to accept as poster as well.